# Micro-Bench: A Vision-Language Benchmark for Microscopy Understanding

**Alejandro Lozano** [*]
Department of Biomedical Data Science
Stanford University
Stanford, CA 94305
lozanoe@stanford.edu

**Jeffrey Nirschl** [*]
Department of Pathology
Stanford University
Stanford, CA 94305
jnirschl@stanford.edu

**James Burgess**
ICME
Stanford University
Stanford, CA 94305
jmhb@stanford.edu

**Sanket Rajan Gupte**
Department of Computer Science
Stanford University
Stanford, CA 94305
sanketg@stanford.edu

**Yuhui Zhang**
Department of Computer Science
Stanford University
Stanford, CA 94305
yuhuiz@stanford.edu

**Alyssa Unell**
Department of Computer Science
Stanford University
Stanford, CA 94305
aunell@stanford.edu

**Serena Yeung-Levy**
Department of Biomedical Data Science
Stanford University
Stanford, CA 94305
syyeung@stanford.edu

## Abstract

Recent advances in microscopy have enabled the rapid generation of terabytes of image data in cell biology and biomedical research. Vision-language models (VLMs) offer a promising solution for large-scale biological image analysis, enhancing researchers' efficiency, identifying new image biomarkers, and accelerating hypothesis generation and scientific discovery. However, there is a lack of standardized, diverse, and large-scale vision-language benchmarks to evaluate VLMs' perception and cognition capabilities in biological image understanding. To address this gap, we introduce Micro-Bench, an expert-curated benchmark encompassing 24 biomedical tasks across various scientific disciplines (biology, pathology), microscopy modalities (electron, fluorescence, light), scales (subcellular, cellular, tissue), and organisms in both normal and abnormal states. We evaluate state-of-the-art biomedical, pathology, and general VLMs on Micro-Bench and find that: i) current models struggle on all categories, even for basic tasks such as distinguishing microscopy modalities; ii) current specialist models fine-tuned on biomedical data often perform worse than generalist models; iii) fine-tuning in specific microscopy domains can cause catastrophic forgetting, eroding prior biomedical knowledge encoded in their base model. iv) weight interpolation between fine-tuned and pre-trained models offers one solution to forgetting and improves general performance across biomedical tasks. We release Micro-Bench under a permissive license [2] to accelerate the research and development of microscopy foundation models.

---

[*]These authors contributed equally.

[2]The dataset is hosted on Hugging Face at: `https://huggingface.co/datasets/jnirschl/uBench`. We publish full code to replicate results at `https://github.com/yeung-lab/Micro-Bench`.

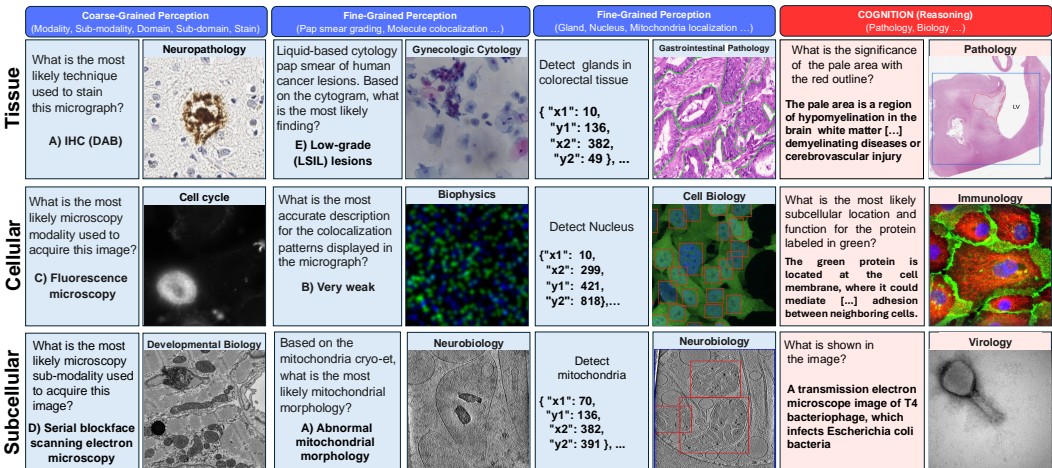

Figure 1: Data samples from Micro-Bench, covering perception (left) and cognition (right) tasks across subcellular, cellular, and tissue levels tasks across electron, fluorescence, and light microscopy.

# 1 Introduction

Microscopy is a cornerstone of biomedical research [19, 54], enabling detailed study of structures at multiple scales [72]. Advances in cryo-electron microscopy, high-throughput fluorescence microscopy, and whole-slide imaging allow scientists to examine and analyze atomic, subcellular, cellular, and tissue-level structures with high precision [21] to reveal new insights into complex biological processes. To achieve this, researchers interpret and contextualize image findings within their existing scientific knowledge to link observations to biological functions and disease relevance [69]. This process requires domain expertise to identify normal and abnormal states, relate observations to molecular and cellular mechanisms, and distinguish artifacts from meaningful findings. Furthermore, learning the nuances of interpreting images across diverse samples or microscopy modalities — beyond an expert's narrow specialization — involves significant trial and error, creating a bottleneck in analyzing the rapidly increasing volumes of imaging data.

Text is an intuitive interface for interactive analysis, and thus vision-language models (VLMs) are one promising approach to assist with image interpretation. A biomedical VLM incorporating knowledge from diverse microscopy images and insights from multiple domain experts could democratize access to scientific knowledge. Such a model could enable text or chat-guided image analysis, help scientists interpret microscopy images outside their field of expertise, facilitate large-scale image annotation, and connect image findings to relevant literature, genes, small molecule therapeutics, and diseases — potentially accelerating scientific research, hypothesis generation, and discovery [23, 70, 64, 52, 57, 12, 46, 46, 41, 28]. To accomplish this, biomedical VLMs must first accurately interpret microscopy data; recognizing basic features such as modality, stains, sample types, and use this information to reason about an image.

However, there is a lack of diverse, large-scale vision-language benchmarks to evaluate image interpretation across multiple microscopy modalities, scales, organisms, and biological states. Existing benchmarks often focus on single-domain diagnostic capabilities (predominantly encompassing histopathology [27, 31]) rather than understanding and describing the fundamental biological mechanisms driving those outcomes . Moreover, in contrast to current trends in the general vision-language community, current biomedical benchmarks lack comprehensive characterization across both perception (e.g., recognizing, localizing, and counting objects) and cognition capabilities (integrating perceptual attributes and knowledge to deduce more complex answers) [22, 44]. This gap hinders tracking the progress and development of robust VLMs tailored for biomedical research.

To address this gap, we present two contributions:

1. **Micro-Bench**: An expert-curated vision-language benchmark comprising 17,235 microscopy images from a diverse collection of 26 published and unpublished datasets, featuring new annotations and a permissive license. Micro-Bench adopts a holistic approach with

three subsets: two perceptual subsets, including five perception coarse-grained tasks (e.g., domain, modality, and stain identification) and 18 perception fine-grained tasks (e.g., cell type classification and segmentation of mitochondria, nucleolus, and glands), along with a cognition component requiring reasoning about images. In total, Micro-Bench includes 24 tasks across light, fluorescence, and electron microscopy (covering 8 submodalities), 25 staining techniques, and 12 scientific domains. Our benchmark is formatted for both closed visual question answering (VQA) and captioning, enabling the evaluation of generative and embedding models .

2. **Characterization** We leverage Micro-Bench to characterize state-of-the-art general and domain-specific biomedical VLMs. We show several findings: even the best-performing VLMs have high error rates across all microscopy tasks (do not generalize well); specialist biomedical VLMs often underperform general VLMs; specialist fine-tuning in specific domains can cause catastrophic forgetting of biological knowledge that existed in the base model; and a simple weight ensembling strategy can mitigate the forgetting problem within Micro-Bench.

## 2 Related Work

**Benchmarking Vision-Language Models** Benchmarks enable the characterization of model behavior, aid the community to better understand AI technology, and influence its development [47]. In the general VLM community, there is an emergence of systematic and holistic model evaluation [79, 44, 78]. To achieve this, VLM benchmark generation typically involves repurposing existing datasets across a collection of multiple tasks [88, 82] and providing a large-scale and standardized evaluation across different tasks [85]. It is usual to organize similar tasks with multiple levels of questions—such as perception and cognition—each further categorized into specific sub-tasks [22]. Once structured, evaluations are typically conducted using one of two approaches: open or VQA. Open VQA can offer more comprehensive insights by simulating the open-ended nature of real-world usage. However, it may require additional experts or models to assess responses, which can introduce potential biases or hallucinations. Alternatively, natural generation metrics may be used, though they often correlate weakly with human responses [20]. Closed VQA mitigates these issues, but requires metadata curated by experts to create meaningful and challenging distractors.

**Biomedical Vision-Language Benchmarks.** While previous works have developed various biomedical vision-language benchmarks that have been instrumental in advancing diagnostic capabilities, they present three main problems: 1) Task simplicity: Most biomedical computer vision benchmarks predominantly focus on narrow classification and segmentation tasks [11, 62, 5]. 2) Lack of diversity: existing vision-language datasets are usually limited to diagnostic imaging such as radiology or pathology [59, 17]; As shown in Table 12, there is a lack of benchmarks for basic research microscopy. 3) Limited Accessibility: While there are large and diverse datasets encompasisng multiple modalities within microscopy, such as PMC-15M [86], , this data is not yet publicly accessible for training or evaluation.

**Vision-Language Models in Biomedicine.** Vision-language models (VLMs) can be generally categorized into two types: 1) Contrastive models, such as CLIP [58] and ALIGN [36], which use contrastive learning to create shared image-text embeddings, facilitating tasks like zero-shot classification and text-image retrieval; and 2) Auto-regressive models, such as Flamingo [4] and GPT-4 [3], which integrate image embeddings with large language models (LLMs) to perform zero-shot tasks, follow instructions, and reason about content.

While vision-language models (VLMs) have significant potential to advance biomedicine [55, 69, 23, 70, 50], they are primarily trained on general datasets with limited biomedical coverage, leading to suboptimal performance on biomedical tasks [63, 89]. Hence, specialized VLMs have been developed by fine-tuning generalist models on biomedical data. Notable examples of contrastive models include BiomedCLIP [86], trained on images from PubMed, and histopathology vision-language models such as PLIP [32] and CONCH [83], which were trained on Twitter (now X) and pathology-specific literature, respectively. Despite the advancements in model development, there is still a lack of comprehensive benchmarks to evaluate the performance of image-based perception and reasoning about microscopy images across diverse scales and modalities, limiting our understanding of failure modes within these domains. Our work addresses this issue by providing a comprehensive benchmark across diverse biological processes, organisms, microscopy modalities, domains, and tasks.

Figure 2: Micro-Bench construction protocol. **Perception dataset (left):** experts taxonomize use cases across subcellular, cellular, and tissue-level applications and collect representative datasets spanning multiple imaging modalities to test those scenarios. Next, datasets are converted to a common format, and the ontological information extracted from their metadata is manually standardized. Aided by this information, experts synthesize VQA pairs designed to test perception ability. **Cognition dataset (right):** First, domain experts use an interactive web application to upload their images and corresponding open-ended VQA pairs. Next, GPT-4 transforms the VQA pairs into a close-ended multiple-choice format. All GPT-4 generations are reviewed by experts before being incorporated into the cognition dataset.

# 3   Dataset collection methodology

Recognizing the need for an expert-level benchmark in microscopy for comprehensive biological and biomedical understanding, we developed a benchmark to assess VLMs' perception and cognition capabilities in microscopy image analysis following the methodology shown in Figure 2. At a high level, the pipeline consists of two main components: (i) Biomedical experts categorized potential tasks and collected diverse microscopy datasets across multiple scientific domains, focusing on evaluating perception capabilities. (ii) We then complement Micro-Bench by crowdsourcing questions from a larger group of expert microscopists using a web application.

## 3.1   Perception Dataset Curation

**Dataset Review and Selection** Open data repositories, including Zenodo, Dataverse, Dryad, and BBBC, among others, were searched for microscopy biomedical image datasets. Data with permissive licenses (CC BY 4.0) allowing derivatives and redistribution were prioritized. A cell biologist and pathologist reviewed the images to ensure high quality (e.g., absence of artifacts or distortion). Diverse datasets were selected to include important biological processes (e.g., cell cycle), organelles (mitochondria, nucleus), and cell/tissue types (e.g., HeLa and HEK cells/cardiac and colorectal tissue). Efforts were made to include diverse biological structures, microscopy modalities, and fields of study. However, basic microscopy research is a broad field, and future work can fill gaps in coverage.

**Standardization** The original datasets had different organizational structures, file formats, and often very little metadata. Information regarding the scientific discipline (domain), microscopy method, staining, pixel calibration, and the organism was manually determined by expert review or consulting the original publication. The base experimental metadata was supplemented with manual annotation of multiple bio-ontology identifiers (SNOMED, BTO, FMA, LOINC, UBERON, etc.) to connect the image data with rich biology concepts and relationships knowledge graphs in the future. All images were converted into lossless PNG files at their original resolution and stored with metadata in a paired JSON file. An MD5 checksum was computed for the image data, and each image was assigned a 128-bit unique identifier. The image-JSON pairs were converted into an Apache Arrow file for public distribution and ease of use through Hugging Face datasets [43].

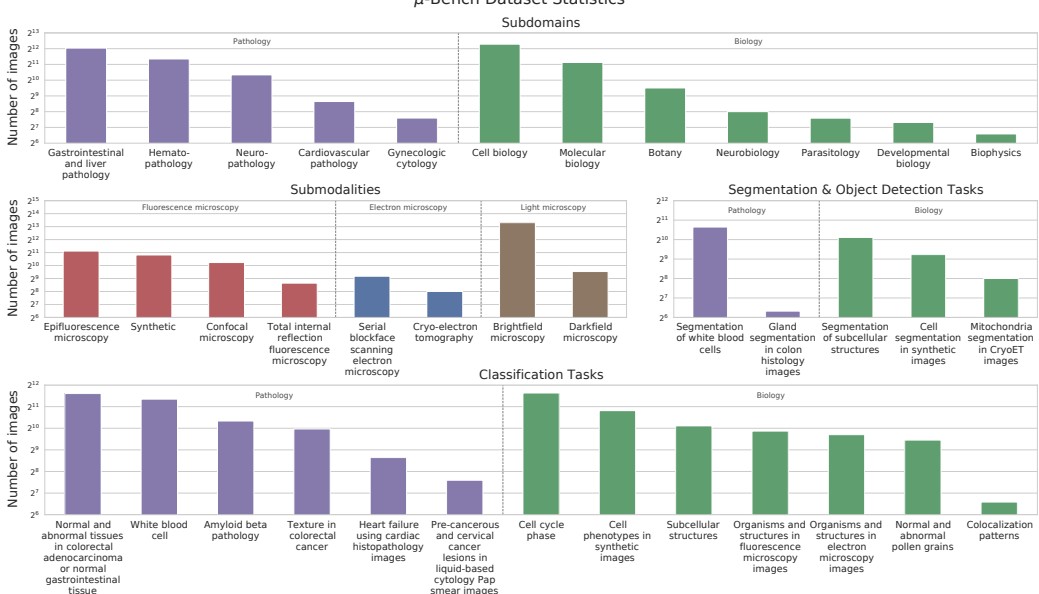

Figure 3: Micro-Bench Perception dataset statistics. The Perception benchmark consists of microscopy images from 12 subdomains in Biology and Pathology, obtained using 8 different imaging techniques, including light, fluorescence, and electron microscopy. It includes 18 perception fine-grained tasks: 13 for classification and 5 for segmentation or object detection.

**VQA task generation** We used the collected standardized metadata to create closed VQA questions that test capabilities at different levels: easier *coarse-grained* perception and challenging *fine-grained* perception (examples are shown in Figure 1).

The coarse-grained perception split tests basic image properties: the broad category of scientific discipline, the type of microscope, or the stain/contrast agent. These groups are visually distinct (e.g., fluorescence vs. electron microscopy) and relatively straightforward even for non-biologists, but provide a framework for image-based reasoning, including expected normal/abnormal findings for a given sample, artifacts specific to a modality, etc. Although easy for humans, these tasks are essential to assess whether VLMs have an intuitive understanding of microscopy images. Furthermore, VLMs with accurate coarse-grained perception can serve as an independent check for the scientist when the text and image input are discordant.

The fine-grained perception split is more challenging. Within each category of a scientific discipline or microscopy modality, there are image classes or features that need to be recognized to perform image interpretation. Dataset-specific tasks include identifying cell type, subcellular organelles, cell cycle phase, and other visually distinct biological processes that are important for reasoning about biological images. Solving fine-grained perception relies on finer-grained visual features and is more challenging for humans.

We formulate both coarse-grained and fine-grained perception as closed VQA. We chose this over open VQA as it's simpler to analyze and doesn't rely on LLMs for automatic evaluation. To generate VQA options in coarse-grained perception, we designed a tree encompassing microscopy modalities (Figure 15), scientific domains (Figure 16), and staining techniques (Figure 17), which enables sampling fine-grained options within concepts (e.g., selecting IHC(DAB) and IHC(RED) as likely stain options for question regarding light microscopy).

**Quality control** Throughout all processing, we validate the schema of each data instance to ensure a consistent format and prevent errors before incorporating into Micro-Bench. The schema includes: *modality*: identification of the microscopy modality (BF, EF, or EM); *submodality*: identification of the microscopy sub-modality (e.g., confocal, phase contrast, or scanning electron microscopy); *domain*: determination of the field of study (e.g., cell biology, histology, or pathology); *subdomain*: identification of the sub-field (e.g., cancer research, neurobiology, or infectious diseases); *staining*:

recognition of the staining technique (e.g., H&E, DAPI, or IHC). We compute the perceptual hash of each image for image deduplication–the number of near duplicates or perceptually similar images in Micro-Bench is no different from comparable biomedical benchmarks E.5. Ten external experts not involved in the study evaluated Micro-Bench; the consensus was that an expert with training could perform the VQA tasks, although the difficulty varied E.3. External experts had diverse backgroundsfig:enter-label, significant microscopy experience (median 9.5 yrs), and included post-docs, two board-certified pathologists, and one PI 10. Additional information can be found in the appendix.

**Localization task generation** We also create a spatial localization benchmark split, which involves predicting the bounding box or segmentation mask for nuclei, mitochondria, cells, and glands(examples are shown in Figure 1). Understanding position and layout enables modeling spatial relationships and context and is fundamental to image understanding. Datasets with segmentation were converted to allow instance segmentation, semantic segmentation, and object detection (bounding box and centroid).

## 3.2  Cognitive Dataset Curation

While perception datasets evaluate the fundamental capabilities of VLMs for microscopy image analysis, they fall short in assessing their ability to use perception to reason about objects. We curated a cognitive dataset to evaluate more advanced aspects like knowledge and reasoning. The cognitive dataset includes questions related to gene pathways, metabolic pathways, cell signaling and signal transduction, cell physiology and function, protein-protein interactions, cell-cell interactions, unique properties of the cell of origin or cell type in the image, cytoskeleton and cell structure or morphology, and drug mechanisms of action. These categories cover fundamental biological concepts and cellular processes to more deeply evaluate VLMs' knowledge in understanding microscopy images.

**Cognition Dataset Collection**  We began by providing detailed guidelines for question creation to experts (see appendix C.4), which ensured consistency and quality across the dataset. Using an internal chat-like web interface, we asked domain experts to submit questions reflecting their daily research activities. We encouraged a focus on questions that required challenging image-based reasoning, domain expertise, interpretation of experimental results, or hypothesis generation.

In addition to submitting questions, experts provided context regarding experimental details, image acquisition, organisms, treatments, and image descriptions. With this comprehensive information, GPT-4V generated answers to the submitted questions. These answers were subsequently reviewed by experts, who evaluated the accuracy and interpretation of the responses.

**Multiple-Choice Question Transformation** The collected pairs (image, question, GPT-4V answer, feedback) were transformed into multiple-choice questions using GPT-4. This transformation was guided by a carefully designed prompt ( appendix E.2), verified by a cell biologist and a pathologist, to ensure the questions are challenging and reflective of real-world scenarios biomedical researchers face. Each transformed question includes an image, a question, and six candidate choices. One choice is correct, while the other five are distractors generated by GPT-4, where one choice is "None of the above." Domain experts verified the validity of the generated questions and manually corrected a small number of questions. Finally, we ensured that correct answers were uniformly distributed among answer choices A to F.

## 4  Dataset Description

**Perception Dataset Statistics** For our perception benchmark, we collected a total of 17,235 microscopy images from 26 distinct public datasets (see Table 5) with permissive licensing, prioritizing open CC-BY licenses. To the best of our knowledge, Micro-Bench Perception is the most diverse microscopy vision-language benchmark, spanning light (LM), fluorescence (FM), and electron microscopy (EM), covering 8 microscopy sub-modalities (see Figure 3), 91 unique cells, tissues, and structures over 25 unique staining techniques (see Figure 6). The perception benchmark subset spans this diversity through closed VQA, object detection, and segmentation.

**Cognition Dataset Statistics** For our cognition benchmark, we collected 54 microscopy images and 121 questions from experts in the field. Entries were received from 6 users across 5 different

institutions. The Micro-Bench Cognition dataset encompasses 3 modalities (fluorescence, electron, light) with 12 sub-modalities, 2 domains (pathology and biology) with 14 sub-domains, and 3 scales (nano, micro, macro), covering a diverse range of topics such as pathology, immunology, and virology. Distributions are shown in Appendix Table 11.

## 5 VLM benchmarking and results

### 5.1 Benchmarking approach

Data artifacts like Micro-Bench enable studying model behavior within specialist domains. Since our benchmark covers a wide range of biomedical tasks, we can, for the first time, compare biomedical perception and cognition capabilities across microscopy imagining modalities. In this section, we show the utility of Micro-Bench by reporting empirical findings on a range of VLMs.

Table 1: Macro-average accuracy (with bootstrap confidence interval) for coarse-grained and fine-grained perception and cognition (reasoning) in Micro-Bench .

| $\mu$-Bench | | | | | |
|---|---|---|---|---|---|
| Perception (Coarse-Grained) | | Perception (Fine-Grained) | | Cognition (Reasoning) | |
| Model | Accuracy ($\pm$ CI) | Model | Accuracy ($\pm$ CI) | Model | Accuracy ($\pm$ CI) |
| GPT-4o | 62.68 ($\pm$ 0.35) | GPT-4o | 51.73 ($\pm$ 0.82) | GPT-4o | 62.00 ($\pm$ 9.00) |
| CogVLM | 52.05 ($\pm$ 0.35) | BiomedCLIP | 34.65 ($\pm$ 0.75) | QwenVLM | 41.00 ($\pm$ 10.00) |
| QwenVLM | 49.85 ($\pm$ 0.35) | CONCH | 33.64 ($\pm$ 0.72) | CogVLM | 41.00 ($\pm$ 10.00) |
| BiomedCLIP | 47.57 ($\pm$ 0.34) | ALIGN | 31.9 ($\pm$ 0.72) | OpenCLIP | 38.33 ($\pm$ 8.33) |
| ALIGN | 40.7 ($\pm$ 0.34) | CLIP | 30.09 ($\pm$ 0.71) | ALIGN | 31.00 ($\pm$ 9.00) |
| OpenCLIP | 36.34 ($\pm$ 0.33) | OpenCLIP | 29.36 ($\pm$ 0.69) | CLIP | 28.00 ($\pm$ 9.00) |
| PaliGemma | 36.29 ($\pm$ 0.33) | CogVLM | 28.18 ($\pm$ 0.70) | PaliGemma | 25.00 ($\pm$ 8.00) |
| CLIP | 35.41 ($\pm$ 0.34) | QuiltNet | 27.85 ($\pm$ 0.69) | BiomedCLIP | 25.00 ($\pm$ 8.00) |
| PLIP | 31.11 ($\pm$ 0.32) | QwenVLM | 27.81 ($\pm$ 0.70) | CONCH | 18.00 ($\pm$ 7.00) |
| CONCH | 27.84 ($\pm$ 0.31) | PLIP | 25.49 ($\pm$ 0.68) | Random | 17.00 ($\pm$ 7.00) |
| QuiltNet | 26.58 ($\pm$ 0.31) | PaliGemma | 21.29 ($\pm$ 0.64) | PLIP | 17.00 ($\pm$ 7.00) |
| Random | 18.34 ($\pm$ 0.27) | Random | 19.13 ($\pm$ 0.60) | QuiltNet | 13.00 ($\pm$ 6.00) |

⁺ ▢ General autoregressive VLMs ▪ General contrastive VLMS ▪ Pathology contrastive VLMS ▪ Biomedical contrastive VLMS.

To this end, we first categorized VLMs into two groups: generalist models trained on natural images and language, and 'specialist' models, fine-tuned on biomedical data. Within generalist models, we also distinguish between contrastive and auto-regressive models.

**Generalist Contrastive (GC) VLMs** We evaluate ALIGN [36], OpenCLIP [14], and CLIP [58] as the canonical contrastive models for natural images. Notably, OpenCLIP and CLIP serve as the initial model weights for numerous finteuned specialist biomedical VLMs.

**Generalist autoregressive (GA) VLMs** We evaluate with GPT-4o [3], a state-of-the-art enterprise VLM. For open-source models, we test CogVLM [71], QwenVLM [6], and PaliGemma [9] for their strong performance on general domain tasks, instruction-following capabilities. Furthemore, QwenVLM and PaliGemma support object detection.

**Specialist contrastive (SC) VLMs** Our specialist model selection included two constraints: choosing the best-performing models and preferring minimal architectural changes from their base generalist versions, allowing performance analysis based on variations in training mixtures. We selected BiomedCLIP [86] since it is a robust model with a training set that covers all biomedical imaging modalities in our benchmark (trained on 15 million image-text pairs collected from PubMedCentral). Additionally, we included three pathology VLMs: PLIP (CLIP trained on H&E) [32], QuiltNet (CLIP trained on H&E and IHC) [34], and CONCH (CoCa trained on H&E and IHC) [49], with training dataset sizes of 208k, 1 million, and 1.2 million, respectively. While CONCH and BiomedCLIP are based on CoCa [83] and OpenCLIP, respectively, they modify the architecture or training strategy.

**Evaluation**   The Closed VQA component of Micro-Bench was evaluated with accuracy, generating confidence intervals (CI) via bootstrap [16] (appendix H.2). Object detection was evaluated for models with object detection capabilities (PaliGemma and QwenVLM) in open VQA format using the GRIT localization metric [25] as adopted by prior works [6].

## 5.2   Results

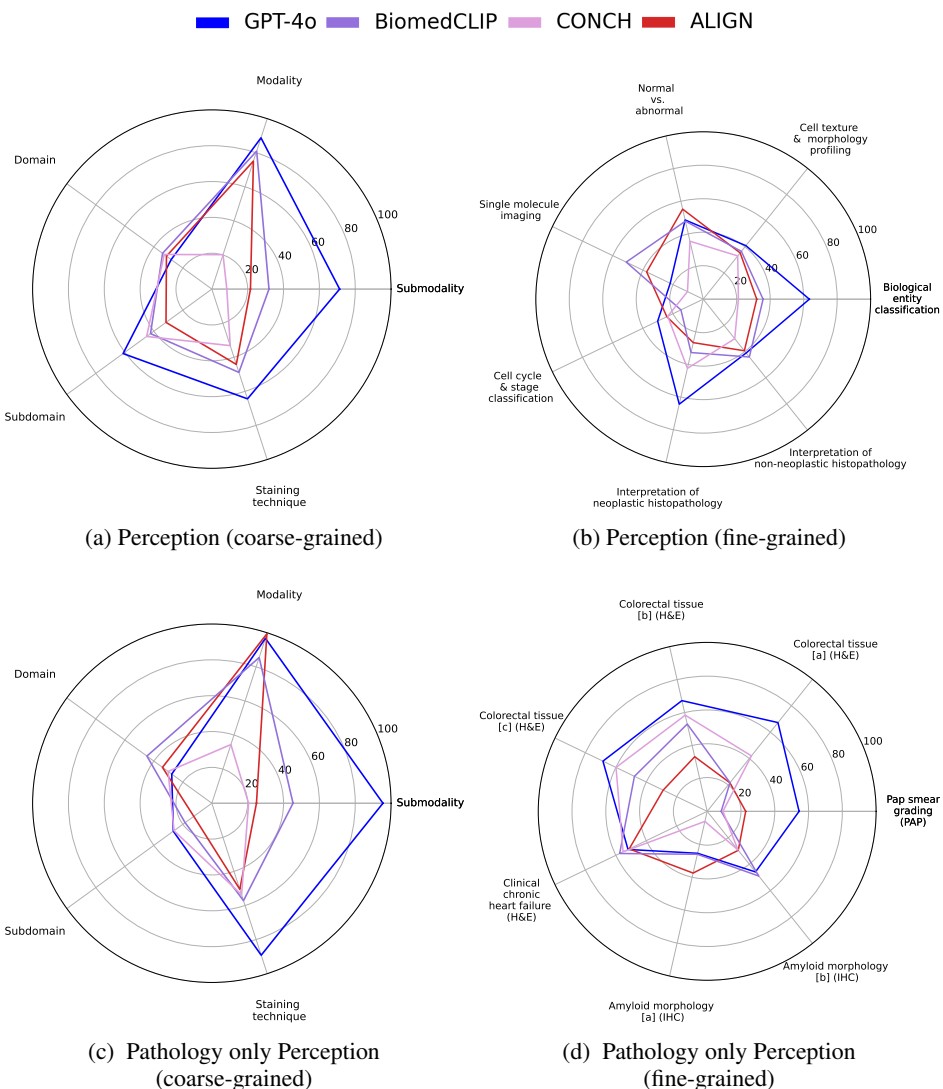

(a) Perception (coarse-grained)

(b) Perception (fine-grained)

(c) Pathology only Perception
(coarse-grained)

(d) Pathology only Perception
(fine-grained)

Figure 4: Performance comparison on the perception benchmark for the best-performing general domain auto-regressive model (GPT-4o), contrastive model (ALIGN), specialist biomedical contrastive model (BiomedCLIP), and specialist pathology contrastive model (CONCH). The top row shows performance in all of the Micro-Bench while the bottom row shows pathology-only samples.

**All models have high error rates**   Table 1 shows the accuracy across perception and cognition. Even the top-performing model (GPT-4o) has high error rates, with an accuracy of 62.6% on coarse-grained perception, 51.7% on fine-grained perception, and 62.0% on cognition tasks. *On average* GPT-4o also outperforms BiomedCLIP (the best biomedical SC VLM) by a minimum of 15% in all evaluation dimensions and CONCH (the best pathology SC VLM) in pathology-specific perception tasks, showing a difference of 39.37% on coarse-grained and 19.40% in fined-grained tasks (as illustrated in Table 1. However, finer subgroup analysis (Figure 11) shows that GPT-4o does not excel in all perception tasks, including domain identification (coarse-grained), single-molecule imaging,

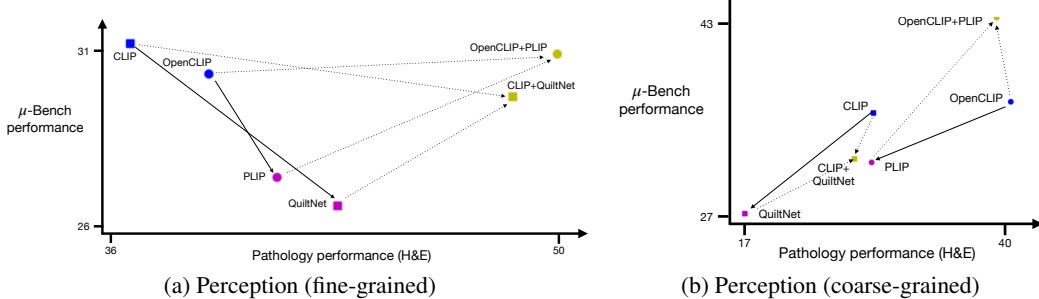

Figure 5: Fine-tuning and microscopy perception generalization on Micro-Bench . Base CLIP models (blue) are fine-tuned to PLIP and QuiltNet using pathology data mixtures (pink). Weight-merging base models with their corresponding fine-tuned models (olive) improves specialist zero-shot performance on Micro-Bench coarse-grained (**Left**) and fined-grained (**Right**) perception.

normal vs abnormal classification, and non-neoplastic histopathology interpretation (fine-grained). The model architecture and training data for GPT-4 are closed source, making it challenging to conclude from these results. However, GPT-4o's high error rates, its substantial gap compared to SC models, and performance variation across task subgroups highlight that $\mu$-Bench is challenging and is not saturated by state-of-the-art general, biomedical, and pathology models.

**Specialist biomedical models are often worse than non-specialist models** While specialist models are explicitly developed for the biomedical domain, they often underperform non-specialized open-source models. For example, in both coarse-grained perception and cognition tasks (Table 1), GA models (CogVLM and QwenVLM) outperform the best SC model (BiomedCLIP) by 4.4% and 16.0% margins respectively. While GA models have a different training objective, larger training mixture, and more model parameters, a similar trend is observed with GC models (ALIGN, OpenCLIP, and CLIP) as they outperform all pathology VLMs in the same tasks by at least 9.5% (PLIP- ALIGN) and 20.3% (CONCH - OpenCLIP) respectively. This ranking is reversed in fine-grained perception tasks, where BiomedCLIP and CONCH perform best. Indeed, fine-grained perception closely resembles the data mixture used to fine-tune contrastive specialist models [31]. This characterization shows weakness in current microscopy biomedical model development, which we investigate next.

**Specialist training can cause catastrophic forgetting** Generalist contrastive models like (OpenCLIP and CLIP) surprisingly outperform their fine-tuned counterparts (PILP and QuiltNet) in coarse-grained perception and cognition (Table 1). Specifically, PILP and QuiltNet are fine-tuned directly from OpenCLIP and CLIP using pathology-only (brightfield microscopy) data closest to Micro-Bench fine-grained perception tasks. Although it improves performance on pathology-specific perception fine-grained tasks (Figure 4), it degrades performance on all other tasks (Table 1).

**Micro-Bench characterization drives robust model development** To address catastrophic forgetting identified in our multi-level evaluation, we ensemble base model weights (OpenCLIP / CLIP) with fine-tuned model weights (PLIP/QuiltNet) to create merged models (PLIP+OpenCLIP / QuiltNet+CLIP), as proposed in [75]. As shown in Figure 5, when comparing merged models to their fine-tuned counterparts, perception performance increases across all of Micro-Bench (y-axis), including pathology-specific tasks (x-axis). This is remarkable given that model merging (or model ensembling) is a simple and training-free strategy. To our knowledge, this is the first application of merging to biomedical embedding models, suggesting many further opportunities to create generalizable biomedical models.

**Micro-Bench supports probing design decisions for biomedical VLMs** We have shown that Micro-Bench offers valuable insights into microscopy biomedical VLMs and hope it encourages further evaluations of design choices. Data diversity is one factor: Table 1 illustrates that BiomedCLIP, trained across all microscopy modalities in Micro-Bench, surpasses specialist models, albeit with a smaller margin for fine-grained tasks compared to CONCH, which uses pathology data. Regarding model architecture and training strategy, generalist autoregressive models (CogVLM and QwenVLM) outperform contrastive models (ALIGN and CLIP) in coarse-grained perception, but the opposite is true for fine-grained perception. For object localization, PaliGemma outperformed QwenVLM (Table

18) on Micro-Bench, though both performed poorly, and no specialist models support detection. Future research could explore prompting strategies, data curation, and new methods to mitigate catastrophic forgetting. One clear opportunity is to fine-tune a base model on multiple tasks or datasets, and merge them all for more a more generalized robust model [74, 61, 24, 60]. A second direction is alternative weight merging strategies [53, 90, 80].

## 6    Conclusion

Benchmarks drive advancements in machine learning by providing a standard to measure progress and allowing researchers to identify weaknesses in current approaches. Thus, the lack of biomedical vision-language benchmarks limits the ability to develop and evaluate specialist VLMs. We address this gap in microscopy by introducing the most diverse collection of microscopy vision-language tasks spanning perception and cognition. We use Micro-Bench to establish, for the first time, the performance of some of the most capable VLMs available and find high error rates of 30%, highlighting room for improvement. We demonstrate how Micro-Bench can be leveraged to generate new insights. Lastly, we share Micro-Bench to enable researchers to measure progress in microscopy foundation models.

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
