# 7   Appendix

## A   Limitations

Micro-Bench is a diverse benchmark for evaluating vision-language models on microscopy image and text data. The dataset is intended for testing purposes only, not for training. The moderate size will allow many academic researchers to assess the performance of models trained on natural images or other biology datasets.

While Micro-Bench covers various biological length scales, microscopy modalities, scientific disciplines, and organisms, not all domains and modalities are equally represented. This partially reflects the usage patterns of the field, with human samples (cell lines or tissues) being more common in biomedical research. Brightfield and fluorescence microscopy images are also more prevalent in Micro-Bench compared to electron microscopy. This means that the results on Micro-Bench may not apply equally well to all model organisms or electron microscopy images. The VLM performance may be lower in these areas due to the data being rare in both natural image datasets and uncommon in biomedical datasets.

We strive for a high-quality dataset and involve cell biologists and pathologists during dataset creation, quality control, and interpretation of results. However, the field of biology is diverse, and no individual or small group can reliably stay up-to-date with all aspects of biological/biomedical research. For example, we included a botany dataset (ICPR 2020 Pollen) to show a commitment to including diverse scientific disciplines, but we currently do not have expert plant biologist contributors.

Micro-Bench represents the first vision-language benchmark for microscopy covering all major biology length scales and modalities. We see Micro-Bench as a living benchmark we intend to grow, although we acknowledge the aforementioned limitations. Future versions will prioritize incorporating new data from diverse organisms and microscopy modalities to improve representation across all length scales, microscopy modalities, and organisms. We will also benefit from community engagement by involving domain experts from diverse fields and obtaining data to balance currently under-represented areas.

## B   Ethical Compliance and Acknowledgements

### B.1   Ethics Statement

**Safe and ethical use of biomedical data:** We prioritize safe and ethical research practices while creating Micro-Bench . All public datasets with patient-derived histopathology images had already been de-identified by the dataset's original authors in compliance with applicable privacy laws and institutional guidelines. The public histopathology image data and metadata were reviewed, and it was determined that it was not possible to identify any individual from the de-identified data. The Stanford Institutional Review Board guidelines were reviewed and discussed, and the use of the images was determined not to be human subjects research.

The Micro-Bench cognition images and questions were voluntarily submitted by a small (<10) number of biology/pathology users alpha-testing a free web chat application. There was no intervention, experiment vs. control group, or research question during the alpha testing. There was no greater risk of using the app than other internet apps. At registration, users agreed to the service terms, which included releasing image and text data under CC-BY-SA-4.0.

**Consent and data usage:** We thank and respect the original dataset authors and use data according to the original copyright and license. Micro-Bench was developed with both academic and commercial research in mind. Many datasets are a version of CC-BY-4.0 to allow both academic and commercial usage. However, some data restricts commercial applications via non-commercial clauses or CC-BY-NC-4.0-related licenses. While creating Micro-Bench we significantly improved the data by performing expert review, quality control/standardization, expert labeling with biomedical ontology codes, and creating multiple-choice VQA questions and captions for each image. When the original dataset license is CC-BY-4.0, we release our Micro-Bench versions under a permissive CC-BY-SA-4.0 to foster a transparent and collaborative benchmark. For the subset with CC-BY-NC-SA-4.0, we respect the original license and release these data under CC-BY-NC-SA-4.0.

**Bias and data diversity:** We recognize that AI models, including VLMs, can perpetuate or exacerbate biases in the training data, and incorporating diverse data may mitigate these biases. Diversity is key to understanding biological processes and how they vary across biological sex, ethnicity, or other factors. When possible, we annotate cell lines with age, sex, ethnicity, and other metadata from Cellosaurus or the Cell Line Ontology. We consciously include diverse microscopy images from multiple institutions across various organisms, modalities, and biological states to ensure Micro-Bench provides an accurate and fair performance assessment. However, images of human samples and brightfield/fluorescence microscopy are more common in the field and thus over-represented in Micro-Bench .

**Potential negative societal impacts:** AI models trained on biomedical data have the potential for far-reaching impacts on society, both positive and negative. Potential negative impacts include biased performance across different demographic groups and reinforcing existing disparities in research and healthcare. We are committed to mitigating these risks by ensuring the dataset's diversity and continuously reviewing the ethical implications of our work. Additionally, we will engage with the broader research community to identify and address any emerging ethical concerns.

We are committed to ongoing improvements of Micro-Bench to prioritize diverse and representative microscopy data. We will review and update Micro-Bench in response to evolving ethical standards and technological advancements.

## B.2  Conflicts of Interest Disclosures

The authors have no conflicts of interest to disclose.

## B.3  Funding/Support

This research was supported by NIH grants (NIH#P30AG066515 to JJN), the Chan Zuckerberg Initiative Neurodegeneration Challenge Pairs Pilot Project to SYL (2020-221724, 5022), the Wu Tsai Knight Initiative Innovation grant (#KIG 102) to SYL, Arc Institute to AL, Quad Fellowship to JB, and NSF Graduate Research Fellowship (#DGE-2146755) and Stanford Graduate Fellowship to AU. SYL is a Chan Zuckerberg Biohub – San Francisco Investigator.

## B.4  Computing resources

One benefit of Micro-Bench is that it is amenable to use by academic labs of all sizes, even those with limited resources. All evaluation tasks could be run on one NVIDIA A6000 (48GB VRAM), except for QWenVL, where we used one A100 (80GB VRAM). Fast inference for Micro-Bench is provided via EVVLM. Experiments were performed in an on-premises university computing environment with 1024 CPU cores.

## B.5  Acknowledgments

We thank all the domain experts for testing the web app and submitting questions that were eventually used in Micro-Bench cognition. We highlight the contributions of Pedro Guedes-Dias, Andrew Moore, and Julian Perez. We appreciate feedback from Josiah Aklilu and Orr Zohar on earlier manuscript versions.

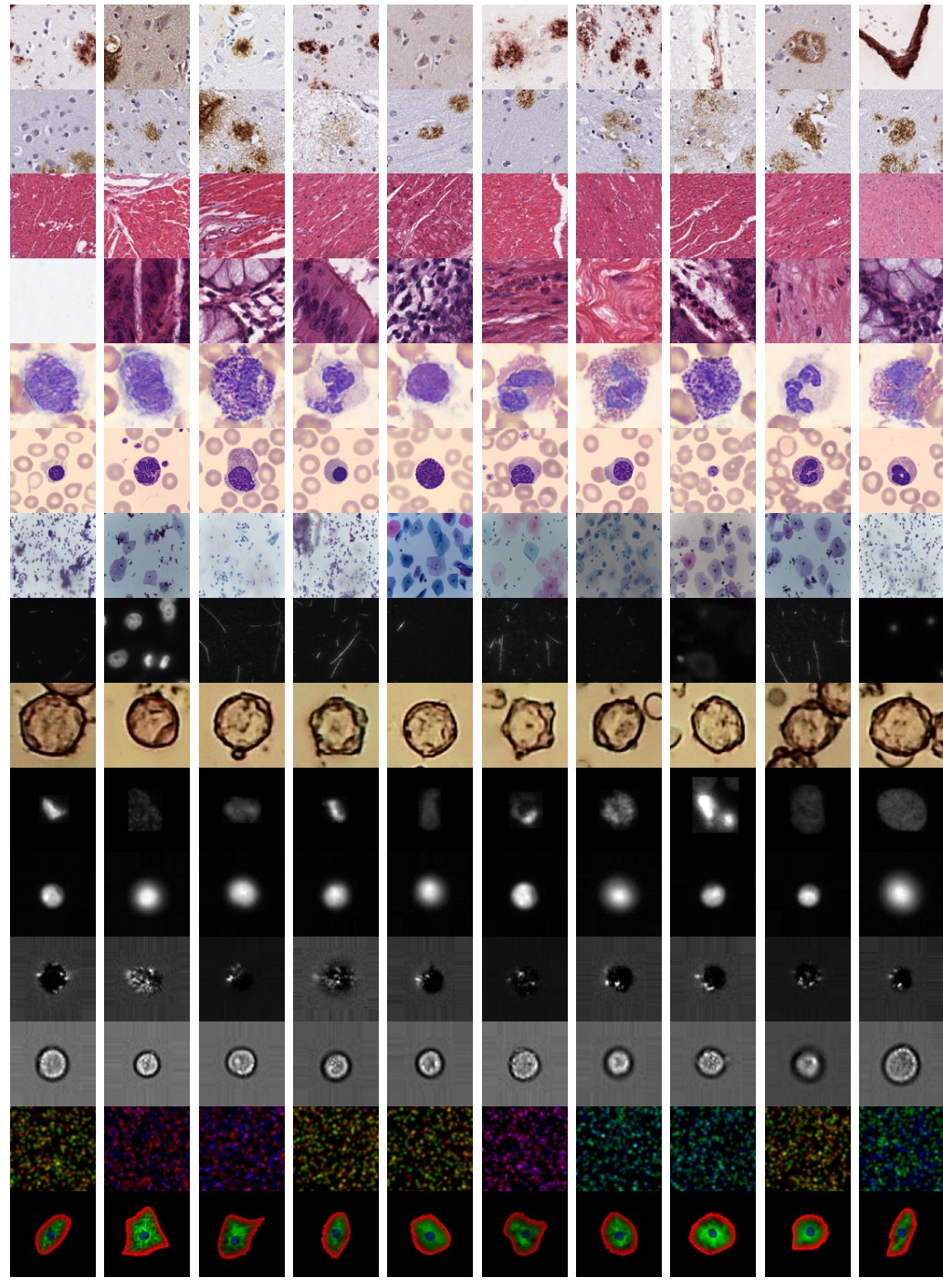

Figure 6: Ten random image samples from each dataset in Micro-Bench [perception].

## C  Benchmark Details

### C.1  Instructions for downloading the benchmark

The benchmark can be downloaded via HuggingFace Datasets at: https://huggingface.co/datasets/jnirschl/uBench.

The HuggingFace Datasets contain multiple questions and captions per data point. Please follow the instructions provided in the official reposiotry (`https://github.com/yeung-lab/Micro-Bench`) to correctly convert the raw dataset into the evaluation benchmark.

### C.2  Micro-Bench overview

Micro-Bench is organized into three main categories:

1. **Micro-Bench Perception (Coarse-grained)**: Containing basic questions about the type of biomedical field of study (domain), subdomain, microscopy modality, submodality, and stain.
2. **Micro-Bench Perception (Fine-grained)**: Identification or questions regarding a biological cell, cellular process, subcellular or tissue structures, biological state (normal/abnormal), etc.
3. **Micro-Bench Cognition (Reasoning)**: Expert-generated questions that typically require visual-based reasoning or integrating knowledge about the micrograph's composition and subject to deduce an answer.
4. **Micro-Bench Object Detection (Localization)**: Bounding box detection of common biological objects, with an easy and hard data split.

Table 3 shows descriptive statistics for Micro-Bench Perception while Table 9 shows statistics for Micro-Bench Cognition. Table 2 shows Micro-Bench demographics across all subsets.

Table 2: Micro-Bench demographics across all subsets group by ethnicity, specimen, and age.

| Concept | Group | Overall | Female | Male | None |
|---|---|---|---|---|---|
| Ethnicity n (%) | African American | 408 (13.8) | 340 (12.6) | 68 (26.6) | |
| | Caucasian | 2477 (84.0) | 2308 (85.7) | 169 (66.0) | |
| | Hispanic | 22 (0.7) | 9 (0.3) | 13 (5.1) | |
| | Other | 41 (1.4) | 35 (1.3) | 6 (2.3) | |
| Specimen n (%) | HEK293 | 1124 (13.5) | 1124 (29.1) | | |
| | HeLa | 319 (3.8) | 319 (8.2) | | |
| | Jurkat E6 | 2229 (26.7) | 2229 (57.6) | | |
| | NHEK | 40 (0.5) | 40 (1.0) | | |
| | Patient | 4564 (54.7) | 144 (3.7) | 256 (83.7) | 4164 (100.0) |
| | U2OS | 13 (0.2) | 13 (0.3) | | |
| | ARPE-19 | 50 (0.6) | | 50 (16.3) | |
| Median age [min,max] (yrs) | | 14.0 [0.0,80.0] | 14.0 [0.0,80.0] | 53.0 [3.0,71.0] | |
| Total | | 17315 | 3869 | 306 | 13140 |

## C.3   Micro-Bench Perception

Table 3 shows descriptive statistics for Micro-Bench Perception, which comprises 17,235 microscopy images collected from 25 unique datasets of 96 different cell and tissue types across light, fluorescence, and electron microscopy. If defined, only images from each dataset's test set are selected; otherwise, a random 15 percent split is created. Each cell/tissue class is sub-sampled to a maximum of 200 sub-classes per class (if the cell type is less than the maximum, the full set is used). Each cell/tissue type is densely annotated (then propagated to each instance of the same type) by an expert, as shown in Figure 20

Table 3: Micro-Bench Perception dataset statistics summary

| Aspect | Count |
|---|---|
| Datasets | 25 |
| Domains | 2 |
| Subdomains | 12 |
| Modalities | 3 |
| Submodalities | 8 |
| Stains | 25 |
| Coarse-grained perception tasks | 5 |
| Fine-grained perception tasks | 18 |
| Classification tasks | 13 |
| Segmentation and Object Detection tasks | 5 |
| Unique Images | 17,235 |

Table 6 provides summary statistics of domain coverage. Overall, the benchmark covers 8,637 biology images and 8,678 pathology images across 12 subdomains. Similarly, Table 7 shows summary statistics of microscopy modalities covered by Micro-Bench perception, including 10,864 images for light microscopy, 5,618 for fluorescence microscopy, and 833 images for electron microscopy across 8 microscopy imaging submodalities and 25 unique microscopy staining techniques (see Table 8).

**Micro-Bench Perception (Coarse-grained)**: Hierarchical metadata for each of the 17,235 perception images and task-specific templates (shown in Table 23) are used to create 5 coarse-grained questions and captions regarding microscopy modality, submodality, domain, subdomain, and staining technique. The use of hierarchical metadata enables the generation of options within each hierarchical level. For example, for microscopy submodality, we leveraged microscopy modality metadata (shown in Figure15 ) to randomly sample submodality options within a microscopy modality (e.g., differential interference contrast microscopy within light microscopy). A total of 86,175 (17,235x5) coarse-grained questions are generated using this approach, leveraging domain metadata Figure 16 as well as staining metadata Figure 17). Section K provides 10 random examples of coarse-grained data points (two per type).

**Micro-Bench Perception (Fine-grained)**: Task-specific metadata for each of the 25 unique classification datasets, along with custom prompts (Table 3), is used to generate 13 unique tasks comprising a total of 17,235 unique question-image pairs in a closed VQA format. Table 4 shows the number of unique images per task.

**Micro-Bench Perception (Object detection)** For datasets with segmentation annotations, we copy the segmentation annotations and also convert them to object detection annotations, giving 3641 images [10, 29, 15, 76, 65].

We define an easy and hard split. The easy split contains the 'cell' class from Burgess et al. [10] dataset and 'nucleus' class from the Held et al. [29]. Here the image contains only the target object, meaning that simple foreground / background segmentation would work well. The hard split contains the 'nucleus' class from Burgess et al. [10], the 'nucleus' class from opencell [15], the 'mitochondria' class from Wu et al. [76], and the 'gland' class from Sirinukunwattana et al. [65]. Here, the target object must be separated from surrounding visual information, and this is challenging.

**Perception dataset forming questions for evaluation** Each sample has a question and set of candidate answers. We describe how to evaluate this VQA task in appendix F.

Table 4: Micro-Bench Perception Fine-grained and Object detection tasks

| Classification Tasks | Images |
|---|---|
| Cell cycle phase | 3169 |
| Normal and abnormal tissues in colorectal adenocarcinoma or normal gastrointestinal tissue | 3114 |
| White blood cell | 2600 |
| Cell phenotypes in synthetic images | 1800 |
| Amyloid beta pathology | 1291 |
| Subcellular structures | 1105 |
| Texture in colorectal cancer | 1000 |
| Organisms and structures in fluorescence microscopy images | 934 |
| Organisms and structures in electron microscopy images | 833 |
| Normal and abnormal pollen grains | 700 |
| Heart failure using cardiac histopathology images | 400 |
| Pre-cancerous and cervical cancer lesions in liquid-based cytology Pap smear images | 193 |
| Colocalization patterns | 96 |
| **Segmentation and Object Detection Tasks** | **Images** |
| Segmentation of white blood cells | 1600 |
| Segmentation of subcellular structures | 1105 |
| Cell segmentation in synthetic images | 600 |
| Mitochondria segmentation in CryoET images | 256 |
| Gland segmentation in benign and malignant colon histology images | 80 |

**Perception dataset source details**: Table 5 lists the dataset names, licenses, and corresponding DOIs or URLs for dataset used. Most of these datasets were sourced from open data repositories, such as Zenodo, Dataverse, Dryad, BBBC, EMPIAR, Kaggle, and various project websites. One unpublished dataset was acquired by one of the authors and released here.

Table 5: Provenance of the datasets for the Micro-Bench dataset. The dataset name is provided along with the original license and URL to the dataset (or DOI where available).

| Dataset | License | Link |
|---|---|---|
| Acevedo et al 2020 [2] | CC-BY-4.0 | DOI |
| PCST-Contour [10] | CC-BY-4.0 | DOI |
| PCST-Eccentricity [10] | CC-BY-4.0 | DOI |
| PCST-Texture [10] | CC-BY-4.0 | DOI |
| Colocalization benchmark [77] | CC-BY-NC-SA-3.0 | DOI |
| EMPIAR SBF-SEM [35] | CC0 | URL, URL, URL, URL, URL |
| BBBC048 (Brightfield) [18] | CC-BY-NC-SA-3.0 | URL |
| BBBC048 (Darkfield) [18] | CC-BY-NC-SA-3.0 | URL |
| BBBC048 (Epifluorescence) [18] | CC-BY-NC-SA-3.0 | URL |
| CellCognition (Golgi) [29] | Attribution | URL |
| CellCognition (H2B) [29] | Attribution | URL |
| CellCognition (Mt) [29] | Attribution | URL |
| Pap Smear 2019 [33] | CC-BY-4.0 | DOI |
| ICPR2020 Pollen [7] | Non-commercial | DOI |
| Jung et al 2022 [37] | CC-BY-NC-4.0 | URL |
| Kather et al 2016 [39] | CC-BY-4.0 | DOI |
| Kather et al 2018 [38] | CC-BY-4.0 | DOI |
| Kather et al 2018 Val7K [38] | CC-BY-4.0 | DOI |
| Nirschl et al 2018 [56] | CC-BY-4.0 | DOI |
| OpenCell [15] | CC-BY-4.0 | URL |
| GlaS Challenge 2015 [65] | Non-commercial | URL |
| Tang et al 2019 [67] | CC-BY-4.0 | DOI |
| Wong et al 2022 [73] | CC-BY-4.0 | DOI |
| Wu et al 2023 [76] | CC-BY-SA-4.0 | DOI |
| Fluorescence Cells & Structures | CC-BY-4.0 | New Data |
| Micro-BenchCognition | CC-BY-4.0 | New Data |

Table 6: Micro-Bench Perception composition by imaging domain and subdomain

| Domains | Images |
|---|---|
| Biology | 8,637 |
| Pathology | 8,678 |

| Subdomains | Images |
|---|---|
| Cell Biology | 4979 |
| Gastrointestinal and Liver Pathology | 4194 |
| Hemato-pathology | 2600 |
| Molecular Biology | 2229 |
| Neuro-pathology | 1291 |
| Botany | 726 |
| Cardiovascular Pathology | 400 |
| Neurobiology | 256 |
| Gynecologic Cytology | 193 |
| Parasitology | 192 |
| Developmental Biology | 159 |
| Biophysics | 96 |

Table 7: Micro-Bench Perception breakdown by imaging modality and submodality

| Modalities | Images |
|---|---|
| Light Microscopy | 10864 |
| Fluorescence Microscopy | 5618 |
| Electron Microscopy | 833 |

| Submodalities | Images |
|---|---|
| Brightfield Microscopy | 10121 |
| Epifluorescence Microscopy | 2217 |
| Synthetic | 1800 |
| Confocal Microscopy | 1201 |
| Darkfield Microscopy | 743 |
| Serial Blockface Scanning Electron Microscopy | 577 |
| Total Internal Reflection Fluorescence Microscopy | 400 |
| Cryo-electron Tomography | 256 |

Table 8: Micro-Bench Perception breakdown by microscopy staining technique.

| Stains | Images |
|---|---|
| H&E | 4594 |
| Giemsa | 2600 |
| Synthetic | 1916 |
| No Stain | 1742 |
| DAPI | 1105 |
| H2B-mCherry | 783 |
| Propidium Iodide | 743 |
| Basic Fuchsin | 700 |
| IHC(DAB) | 610 |
| Uranyl Acetate | 577 |
| IHC(HDab) | 491 |
| AlexaFluor-tubulin | 400 |
| Papanicolaou | 193 |
| IHC(Red) | 190 |
| Hoechst 33342 | 183 |
| GalT–EGFP | 157 |
| CellMask | 125 |
| Tetraspeck Beads | 51 |
| Lysotracker | 46 |
| Fluorescent Beads | 26 |
| Phal | 25 |
| GalT-GFP | 24 |
| Soluble GFP | 21 |
| Uniform Test Slide | 11 |
| LifeAct | 2 |

## C.4 Micro-Bench Cognition

### Cognition dataset: generation details

Table 9 shows statistics for Micro-Bench Cognition. The cognition subset was collected during alpha-testing of a web application chat interface. A group of 6 biology and pathology experts were invited to interact with a web application as they wished during their daily routines (invitation is shown in Figure 22). There was no specific research question, intervention, or experimental vs. control group; this was a free service. User registration was voluntary and required reviewing and accepting the terms of service. The terms of service discussed that this was not a research study and that the risk of harm is insignificant and would be similar to any VLM chatbot. The terms of service indicated that uploading images indicated the user had copyright or permission, that images did not contain offensive content, and that the images and text could be released with a CC-BY-SA-4.0 license.

Table 9: Micro-Bench Cognition dataset statistics summary

| Aspect | Count |
|---|---|
| Submitters | 6 |
| Institutions | 5 |
| Domains | 2 |
| Subdomains | 10 |
| Modalities | 3 |
| Submodalities | 12 |

Figure 7) shows a screenshot of the web application interface. A typical usage involved uploading a microscopy image, providing context about the image (experiment details), and asking a question. The web app processed the submission using GPT-4V and provided an answer in real-time. Users were encouraged to provide feedback on the answer and a correct answer (if known). Users were encouraged to ask questions that required complex visual reasoning, advanced biomedical knowledge, or could be considered challenging for humans. Random samples for cognition questions are shown in Section M

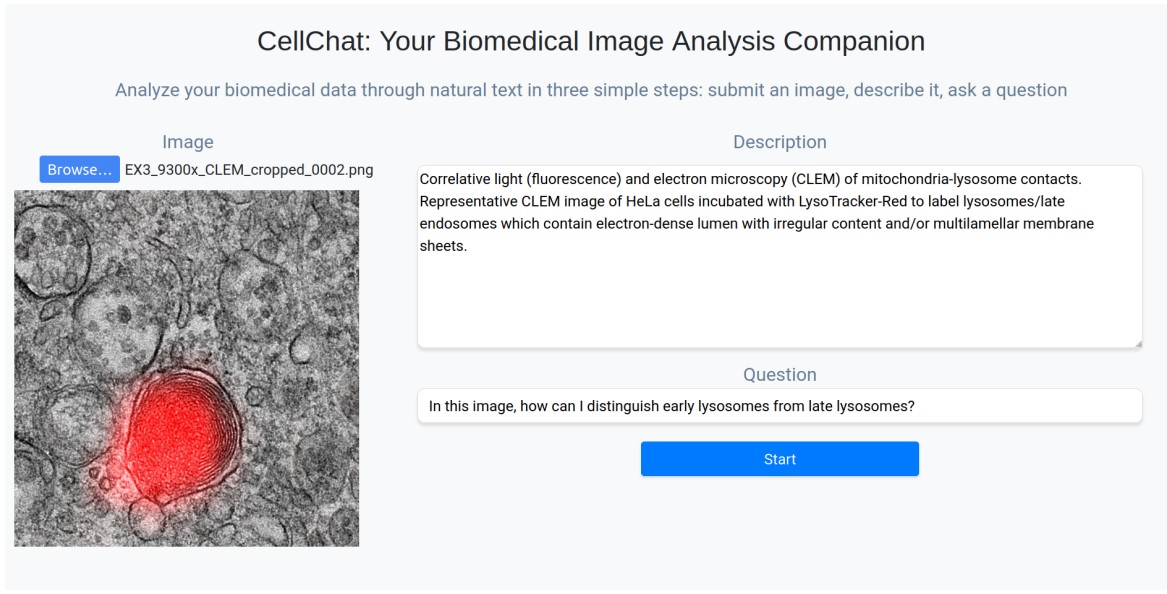

Figure 7: CellChat App: Interactive web app used to elicit real-world open VQA examples from expert microscopists.

Table 10: Micro-Bench Cognition composition by imaging modality and submodality

| Modalities | Questions |
|---|---|
| Fluorescence Microscopy | 76 |
| Light Microscopy | 26 |
| Electron Microscopy | 17 |

| Submodalities | Questions |
|---|---|
| Epifluorescence microscopy | 37 |
| Confocal microscopy | 36 |
| Brightfield microscopy | 21 |
| Cryo-electron tomography | 8 |
| Scanning electron microscopy | 4 |
| Transmission electron microscopy | 4 |
| Differential interference contrast microscopy | 4 |
| Mixed | 3 |
| Transmission electron microscopy (TEM) | 1 |
| Lattice light sheet | 1 |
| Synthetic | 1 |
| Lattice light-sheet microscopy | 1 |

Table 11: Micro-Bench Cognition composition by imaging domain and subdomain

| Domains | Question |
|---|---|
| Biology | 113 |
| Pathology | 8 |

| Subdomains | Question |
|---|---|
| Cell Biology | 45 |
| Cell and molecular biology | 28 |
| Neurobiology | 17 |
| Developmental biology | 8 |
| Immunology | 6 |
| Neuropathology | 6 |
| Gastrointestinal pathology | 2 |
| Virology | 2 |
| Botany | 2 |
| Genetics | 2 |

# D   Comparison to Current Biomedical VQA datasets

Micro-Bench is the first VQA benchmark to encompass all major microscopy modalities. It is also the first biomedical benchmark to include both caption-like and question-like formats, enabling the evaluation of both embedding-based and autoregressive VLMs. Table 12 compares Micro-Bench to current biomedical VQA datasets (closed and open) at the time of publication.

| Dataset | Domains | Source | Unique Images | QA Pairs | Creation | VQA Answer Type | VQA | SEG | CAP | CLS | OD |
|---|---|---|---|---|---|---|---|---|---|---|---|
| **Radiology** | | | | | | | | | | | |
| VQA-Med-2018 [26] | 1 | PubMed Central® | 2,866 | 6,413 | Automated | Open/Close | ✓ | | | | |
| VQA-Med-2019 [1] | 1 | MedPix® | 4,200 | 15,292 | Automated | Open/Close | ✓ | | | | |
| VQA-Med-2020 [8] | 1 | MedPix® | 5,000 | 5,000 | Automated | Open/Close | ✓ | | | | |
| VQA-Med-2021 [66] | 1 | MedPix® | 5000 | 5000 | Automated | Open/Close | ✓ | | | | |
| VQA-RAD [42] | 1 | MedPix® | 315 | 3,515 | Manual | Open/Close | ✓ | | | | |
| RadVisDial (S) [40] | 1 | MIMIC-CXR | 91,060 | 455,300 | Automated | Close | ✓ | | | | |
| RadVisDial (G) [42] | 1 | MIMIC-CXR | 100 | 500 | Manual | Close | ✓ | | | | |
| OVQA [30] | 1 | EMRs | 2,001 | 19,020 | Automated | Open/Close | ✓ | | | | |
| SLAKE [48] | 1 | Decathlon,NIH Chest X-ray, CHAOS | 642 | 15,00 | Manual | Open/Close | ✓ | ✓ | | | |
| **Pathology** | | | | | | | | | | | |
| PathVQA [27] | 1 | PEIR Digital Library | 4,998 | 32,799 | Automated | Open | ✓ | | | | |
| OpenPath [32] | 1 | Twitter (Now X) | 208,414 | 208,414 | Automated | | | | | ✓ | |
| **Biomedical** | | | | | | | | | | | |
| PMC-VQA [87] | 5+* | PubMed Central® | 149,000 | 227,000 | Automated | Close | ✓ | | | | |
| **Biology** | | | | | | | | | | | |
| Multimodality Cell Segmentation Challenge [51] | 1 | 20 Laboratories | 1,500 | - | | | | | | ✓ | |
| Micro-Bench (ours) | 3 * | Curated Datasets | 17,356 | | Expert guided | Open/Close | ✓ | ✓ | ✓ | ✓ | ✓ |

Table 12: Comparison of Micro-Bench to existing **composite** medical and biomedical datasets. Only publicly available datasets were considered. 5+*: Mostly Radiology, Pathology, Microscopy, Signals and Generic biomedical illustrations.

# E   Benchmark Difficulty

## E.1   Comparison with a supervised linear model trained on DINOv2 Features

Although some datasets have been used in the biomedical computer vision community [39, 38], others do not have well-established performance baselines. Thus, we assess the solvability and difficulty of each task via three experiments.

1. We evaluate a supervised linear model's baseline classification performance for the fine-grained and coarse-grained perception tasks using DinoV2-features, demonstrating the ability to discriminate among classes with high accuracy using visual features.

2. We surveyed ten external microscopy experts (including clinicians and scientists) across six domains to assess task feasibility and difficulty, with the expert reporting that the tasks vary in difficulty, but all could be solved with training or expertise.

3. We measure image perceptual similarity and find, as shown in Tables 1 and 2, that the average weighted perceptual similarity of the Micro-Bench benchmark is comparable to other influential biomedical datasets and benchmarks used in hundreds of studies.

Details are shared below:

## E.2 Comparison with a supervised linear model trained on DINOv2 Features

We evaluate a supervised linear model's baseline classification performance for the fine-grained and coarse-grained perception tasks. This establishes that the questions in our benchmark are solvable using image features.

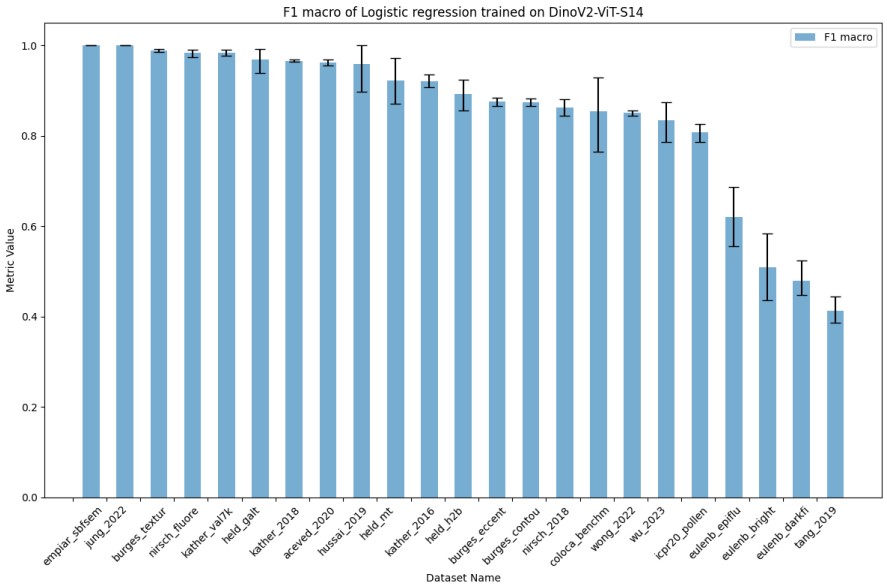

Figure 8: F1 Macro results for logistic regression classifier trained using DinoV2 features.

The images in Micro-Bench represent the test subset of the original data. The original data were split into train, validation, and test subsets according to the published splits. If no published splits were available, which was the case for many previously unused datasets, we created train/val/test splits (0.7/0.1/0.2) stratifying samples by the class label. We use DINOv2 S14 features (384 dims) because they are robust and can train strong linear models on many natural image classification datasets. We extract DinoV2-S14 features from the train/val subsets and train a logistic regression classifier to determine the performance of a linear classifier for these tasks. We use PCA dimensionality reduction (0.95% var) and whitening on the features with Optuna for hyperparameter tuning. Baseline performance was determined using the best dummy classifier, predicting based on a random sampling of prior probabilities, the most common class, etc.

Figure 8 illustrates that most datasets provide sufficient signal to train a robust linear classifier with DinoV2 feature representations primarily learned from natural images. Our results show a weighted average accuracy of 0.86 across all classification tasks. The lowest performance was observed in the classification of mitosis stages in darkfield microscopy (50.96%). In contrast, the highest performance was seen in the classification of synthetic white blood cells in bright microscopy and the organisms and structures in electron microscopy, with a weighted average accuracy of 100%. Lastly, 58.33 % of the tasks achieved a balanced accuracy greater than 80%. These results show a weighted average of 86.71% across all classification tasks with the upper and lower performance range we could expect from a VLM.

## E.3 Benchmark Difficulty Assessment Protocol

We surveyed ten external microscopy experts (both clinicians and scientists) across six domains to assess task feasibility and task difficulty (please refer to section E.4 for more information). To this end, we designed a Google Form to gather expert evaluations. Random samples representative of each class were presented to experts. Subsequently, they were prompted to respond to the following question: *"With sufficient training, could a human expert reliably distinguish between classes in task?"* If so, they were then asked to rate the difficulty on a scale of 1 to 5: *"How difficult is it to distinguish between classes?"*

The difficulty scale was defined as follows:

1. Very Easy
2. Easy
3. Challenging, but doable
4. Hard
5. Very Difficult

### E.4 Benchmark Difficulty Evaluators Statistics

Ten expert scientists participated in this assessment, all of whom were external to the study and did not contribute to its authorship. These experts have an average of nine years of experience (ranging from 3 to 17 years) across diverse fields, including biology, microbiology, immunology, cell biology, and pathology (Figure 9). They also represent various stages of scientific careers and clinical practice (Figure 10) ranging from PhD Students to Principal Investigators. For each task, the majority of experts (at least 8 out of 10) confirmed that **all tasks were feasible for human experts**. Table 13 summarizes the experts' perceived difficulty ratings for distinguishing between classes in each task.

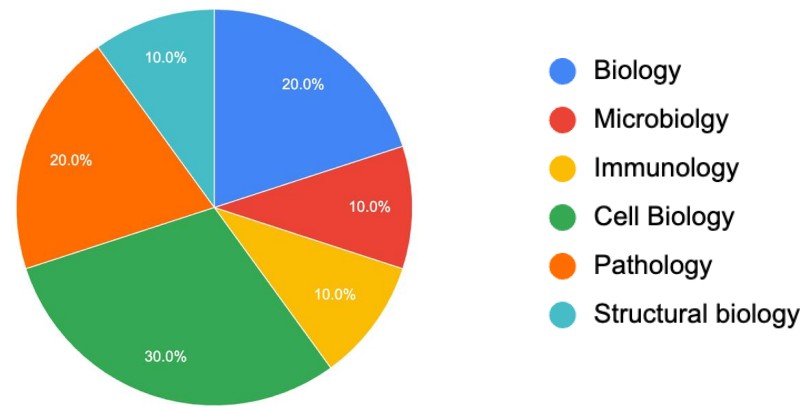

Figure 9: Fields of study of experts evaluating the perceived difficulty of Micro-Bench

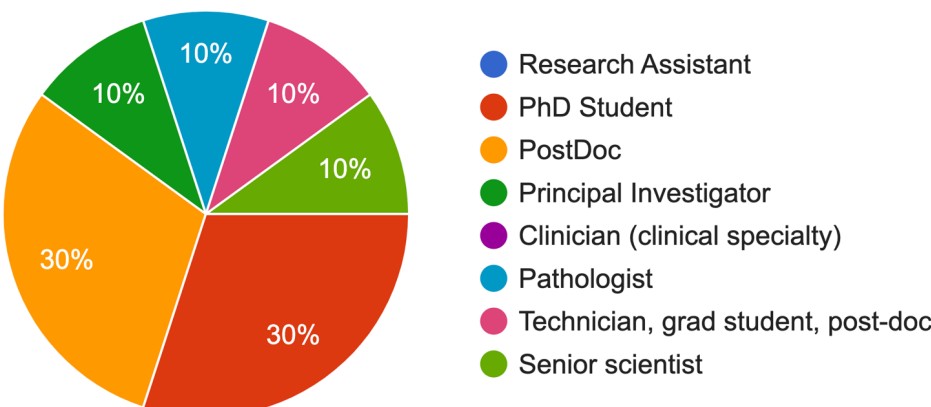

Figure 10: Distribution of experts evaluating the perceived difficulty of Micro-Bench

### E.5 Perceptually similar images

We identify images that are perceptually similar using the well-established perceptual hash algorithm [84]. The method operates in the frequency domain and applies the discrete cosine transform to the

Table 13: Expert Perceived difficulty for perception Subset: Difficulty assigned to each task within perception next to linear probe performance on each task

| Dataset | Linear Probe Macro F1 | Mean Human-perceived difficulty | Median human-perceived Difficulty |
|---|---|---|---|
| **Perception (Coarse-grained)** | | | |
| Stain | 98.59 | 1.71 | 1 |
| Modality | 100 | 1.57 | 1 |
| SubModality | 99.86 | 2.43 | 3 |
| Domain | 100 | 2.14 | 2 |
| SubDomain | 99.82 | 3.43 | 4 |
| | | | |
| Macro Average | 99.65 | 2.25 | 2.2 |
| Weighted Average | 99.65 | 2.25 | 2.2 |
| Min | 98.59 | 1.57 | 1 |
| Max | 100 | 3.42 | 4 |
| | | | |
| **Perception (Fine-grained)** | | | |
| eulenberg_et_al_2017_epifluorescence | 61.96 | 3.86 | 4 |
| eulenberg_et_al_2017_darkfield | 47.99 | 3.57 | 4 |
| held_et_al_2010_h2b | 89.2 | 2.57 | 2 |
| empiar_sbfsem | 100 | 1.71 | 2 |
| burgess_et_al_2024_contour | 87.44 | 3.14 | 3 |
| wong_et_al_2022 | 85.06 | 2.43 | 2 |
| eulenberg_et_al_2017_brightfield | 47.99 | 2.00 | 2 |
| held_et_al_2010_mt | 92.24 | 3.29 | 3 |
| wu_et_al_2023 | 83.42 | 3.29 | 4 |
| colocalization_benchmark | 85.43 | 2.71 | 2 |
| held_et_al_2010_galt | 96.85 | 3.57 | 4 |
| burgess_et_al_2024_eccentricity | 87.54 | 3.57 | 4 |
| burgess_et_al_2024_texture | 98.92 | 3.57 | 4 |
| tang_et_al_2019 | 41.23 | 1.86 | 2 |
| kather_et_al_2018 | 96.64 | 2.43 | 3 |
| nirschl_unpub_fluorescence | 98.37 | 2.14 | 2 |
| opencell | 56.7 | 2.14 | 2 |
| acevedo_et_al_2020 | 96.28 | 1.71 | 2 |
| kather_et_al_2018_val7k | 98.37 | 2.43 | 3 |
| nirschl_et_al_2018 | 86.35 | 2.43 | 2 |
| sirinukunwattana_et_al_2016 | 96.27 | 2.43 | 2 |
| hussain_et_al_2019 | 95.89 | 2.43 | 3 |
| kather_et_al_2016 | 92.15 | 2.57 | 3 |
| jung_et_al_2022 | 100 | 2.29 | 2 |
| icpr2020_pollen | 80.76 | 4.00 | 4 |
| | | | |
| Macro Average | 84.12 | 2.73 | 2.8 |
| Weighted Average | 84.54 | 2.58 | 2.73 |
| Min | 41.23 | 1.57 | 2 |
| Max | 100 | 4 | 4 |

grayscale image to produce a 64-bit binary hash by comparing each DCT coefficient to the median value, which allows for robust identification of perceptually similar images [84]. The pHash algorithm has been widely used in image retrieval and similarity searches for over 10 years (Longjiang et al. 2006, Tang et al. 2014, and Li et al. 2018), and is still used in certain cases of image search and retrieval or copyright infringement, among other use cases. We use a Hamming distance of $\leq 5$ as a threshold for similarity.

Table 14 displays the perceptual similarity across all datasets. Only "Eulenberg et al. 2017 epifluorescence" and "Eulenberg et al. 2017 darkfield" have similarities above 40%. The weighted average similarity for all datasets is 13.43%. For comparison, the MedMNIST dataset [81], detailed in Table 2, shows a similar weighted average perceptual similarity to $\mu$Bench.

To visually investigate the dataset heterogeneity, we embedded each dataset using DINOv2 S14 and applied UMAP for dimensionality reduction. The UMAP projections of each dataset are presented at the end of this file.

We then filtered the benchmark by removing datasets with a similarity percentage above 50%. The impact of removing datasets with high perceptual similarity from our benchmark is reported in Table

4. The average absolute difference in performance is 1.2%. We highlight that the rankings remain unchanged.

Table 14: $\mu$Bench Ranked inter dataset perceptual similarity

| Dataset | Perceptual Similar Images | Total Images | Percent Similar |
|---|---|---|---|
| eulenberg_et_al_2017_epifluorescence | 663 | 743 | 89.23 |
| eulenberg_et_al_2017_darkfield | 434 | 743 | 58.41 |
| held_et_al_2010_h2b | 236 | 646 | 36.53 |
| empiar_sbfsem | 209 | 577 | 36.22 |
| burgess_et_al_2024_contour | 201 | 600 | 33.50 |
| wong_et_al_2022 | 231 | 800 | 28.88 |
| eulenberg_et_al_2017_brightfield | 200 | 743 | 26.92 |
| held_et_al_2010_mt | 29 | 137 | 21.17 |
| wu_et_al_2023 | 39 | 256 | 15.23 |
| colocalization_benchmark | 10 | 96 | 10.42 |
| held_et_al_2010_galt | 10 | 157 | 6.37 |
| burgess_et_al_2024_eccentricity | 27 | 600 | 4.50 |
| burgess_et_al_2024_texture | 24 | 600 | 4.00 |
| tang_et_al_2019 | 7 | 491 | 1.43 |
| kather_et_al_2018 | 5 | 1800 | 0.28 |
| nirschl_unpub_fluorescence | 2 | 934 | 0.21 |
| opencell | 0 | 1105 | 0.00 |
| acevedo_et_al_2020 | 0 | 1600 | 0.00 |
| kather_et_al_2018_val7k | 0 | 1314 | 0.00 |
| nirschl_et_al_2018 | 0 | 400 | 0.00 |
| sirinukunwattana_et_al_2016 | 0 | 80 | 0.00 |
| hussain_et_al_2019 | 0 | 193 | 0.00 |
| kather_et_al_2016 | 0 | 1000 | 0.00 |
| jung_et_al_2022 | 0 | 1000 | 0.00 |
| icpr2020_pollen | 0 | 700 | 0.00 |
| **Weighted Average** | | | 13.43 |
| **Macro Average** | | | 14.93 |

Table 15: Table 2: MedMnist Ranked inter dataset perceptual similarity

| Split | Perceptual Similar Images | Total Images | Percent Similar |
|---|---|---|---|
| organamnist_224 | 11530 | 17778 | 64.85 |
| organamnist_64 | 11527 | 17777 | 64.84 |
| organamnist_128 | 11517 | 17778 | 64.78 |
| organamnist_28 | 9926 | 17778 | 55.83 |
| organsmnist_224 | 1110 | 8827 | 12.57 |
| organsmnist_128 | 1101 | 8827 | 12.47 |
| organsmnist_64 | 1096 | 8827 | 12.41 |
| chestmnist_128 | 2644 | 22433 | 11.78 |
| chestmnist_64 | 2617 | 22433 | 11.66 |
| Chestmnist_28 | 2607 | 22433 | 11.62 |
| chestmnist_224 | 2606 | 22433 | 11.61 |
| organcmnist_64 | 938 | 8216 | 11.41 |
| organcmnist_128 | 926 | 8216 | 11.27 |
| organcmnist_224 | 926 | 8216 | 11.27 |
| retinamnist_128 | 40 | 400 | 10 |
| retinamnist_224 | 40 | 400 | 10 |
| retinamnist_64 | 40 | 400 | 10 |
| Retinamnist_28 | 37 | 400 | 9.25 |
| organsmnist | 673 | 8827 | 7.62 |
| organcmnist | 582 | 8216 | 7.0 |
| octmnist | 52 | 1000 | 5.2 |
| octmnist_128 | 50 | 1000 | 5 |
| octmnist_224 | 50 | 1000 | 5 |
| octmnist_64 | 50 | 1000 | 5 |
| breastmnist_128 | 4 | 156 | 2.56 |
| breastmnist_224 | 4 | 156 | 2.56 |
| Breastmnist_28 | 4 | 156 | 2.56 |
| breastmnist_64 | 4 | 156 | 2.56 |
| pneumoniamnist_128 | 14 | 624 | 2.24 |
| pneumoniamnist_224 | 14 | 624 | 2.24 |
| pneumoniamnist_64 | 12 | 624 | 1.92 |
| Pneumoniamnist_28 | 12 | 624 | 1.92 |
| dermamnist_128 | 24 | 2005 | 1.19 |
| dermamnist_224 | 24 | 2005 | 1.19 |
| Dermamnist_28 | 23 | 2005 | 1.14 |
| dermamnist_64 | 21 | 2005 | 1.04 |
| tissuemnist_224 | 207 | 47280 | 0.43 |
| tissuemnist_64 | 200 | 47280 | 0.42 |
| Tissuemnist_28 | 200 | 47280 | 0.42 |
| tissuemnist_128 | 197 | 47280 | 0.41 |
| pathmnist_224 | 4 | 7180 | 0.055 |
| pathmnist_128 | 2 | 7180 | 0.027 |
| bloodmnist_128 | 0 | 3421 | 0 |
| bloodmnist_224 | 0 | 3421 | 0 |
| bloodmnist_64 | 0 | 3421 | 0 |
| Bloodmnist_28 | 0 | 3421 | 0 |
| pathmnist_64 | 0 | 7180 | 0 |
| Pathmnist_28 | 0 | 7180 | 0 |
| | | | |
| **Weighted Average** | | | 13.37 |
| **Macro Average** | | | 9.74 |

Table 16: Table 1: $\mu$Bench Ranked inter dataset perceptual similarity. The original authors of Eulenberg et al. 2017 (brightfield, darkfield, epifluorescence) processed the image data with border reflection to obtain the desired size. Hence, we expect increased perceptual similarity for the Eulenberg datasets. The raw data in Held et al. 2010 had irregular sizes and shapes. As a part of our processing pipeline, we centered the images on a black background, and thus, we expect some of the Held datasets to have a higher perceptual similarity.

| Dataset | Perceptual Similar Images | Total Images | Percent Similar |
|---|---|---|---|
| eulenberg_et_al_2017_epifluorescence | 663 | 743 | 89.23 |
| eulenberg_et_al_2017_darkfield | 434 | 743 | 58.41 |
| held_et_al_2010_h2b | 236 | 646 | 36.53 |
| empiar_sbfsem | 209 | 577 | 36.22 |
| burgess_et_al_2024_contour | 201 | 600 | 33.50 |
| wong_et_al_2022 | 231 | 800 | 28.88 |
| eulenberg_et_al_2017_brightfield | 200 | 743 | 26.92 |
| held_et_al_2010_mt | 29 | 137 | 21.17 |
| wu_et_al_2023 | 39 | 256 | 15.23 |
| colocalization_benchmark | 10 | 96 | 10.42 |
| held_et_al_2010_galt | 10 | 157 | 6.37 |
| burgess_et_al_2024_eccentricity | 27 | 600 | 4.50 |
| burgess_et_al_2024_texture | 24 | 600 | 4.00 |
| tang_et_al_2019 | 7 | 491 | 1.43 |
| kather_et_al_2018 | 5 | 1800 | 0.28 |
| nirschl_unpub_fluorescence | 2 | 934 | 0.21 |
| opencell | 0 | 1105 | 0.00 |
| acevedo_et_al_2020 | 0 | 1600 | 0.00 |
| kather_et_al_2018_val7k | 0 | 1314 | 0.00 |
| nirschl_et_al_2018 | 0 | 400 | 0.00 |
| sirinukunwattana_et_al_2016 | 0 | 80 | 0.00 |
| hussain_et_al_2019 | 0 | 193 | 0.00 |
| kather_et_al_2016 | 0 | 1000 | 0.00 |
| jung_et_al_2022 | 0 | 1000 | 0.00 |
| icpr2020_pollen | 0 | 700 | 0.00 |
| **Weighted Average** | | | 13.43 |
| **Macro Average** | | | 14.93 |

Table 17: Table 2: MedMNIST Ranked inter dataset perceptual similarity

| Split | Perceptual Similar Images | Total Images | Percent Similar |
|---|---|---|---|
| organamnist_224 | 11530 | 17778 | 64.85 |
| organamnist_64 | 11527 | 17777 | 64.84 |
| organamnist_128 | 11517 | 17778 | 64.78 |
| organamnist_28 | 9926 | 17778 | 55.83 |
| organsmnist_224 | 1110 | 8827 | 12.57 |
| organsmnist_128 | 1101 | 8827 | 12.47 |
| organsmnist_64 | 1096 | 8827 | 12.41 |
| chestmnist_128 | 2644 | 22433 | 11.78 |
| chestmnist_64 | 2617 | 22433 | 11.66 |
| Chestmnist_28 | 2607 | 22433 | 11.62 |
| chestmnist_224 | 2606 | 22433 | 11.61 |
| organcmnist_64 | 938 | 8216 | 11.41 |
| organcmnist_128 | 926 | 8216 | 11.27 |
| organcmnist_224 | 926 | 8216 | 11.27 |
| retinamnist_128 | 40 | 400 | 10 |
| retinamnist_224 | 40 | 400 | 10 |
| retinamnist_64 | 40 | 400 | 10 |
| Retinamnist_28 | 37 | 400 | 9.25 |
| organsmnist | 673 | 8827 | 7.62 |
| organcmnist | 582 | 8216 | 7.0 |
| octmnist | 52 | 1000 | 5.2 |
| octmnist_128 | 50 | 1000 | 5 |
| octmnist_224 | 50 | 1000 | 5 |
| octmnist_64 | 50 | 1000 | 5 |
| breastmnist_128 | 4 | 156 | 2.56 |
| breastmnist_224 | 4 | 156 | 2.56 |
| Breastmnist_28 | 4 | 156 | 2.56 |
| breastmnist_64 | 4 | 156 | 2.56 |
| pneumoniamnist_128 | 14 | 624 | 2.24 |
| pneumoniamnist_224 | 14 | 624 | 2.24 |
| pneumoniamnist_64 | 12 | 624 | 1.92 |
| Pneumoniamnist_28 | 12 | 624 | 1.92 |
| dermamnist_128 | 24 | 2005 | 1.19 |
| dermamnist_224 | 24 | 2005 | 1.19 |
| Dermamnist_28 | 23 | 2005 | 1.14 |
| dermamnist_64 | 21 | 2005 | 1.04 |
| tissuemnist_224 | 207 | 47280 | 0.43 |
| tissuemnist_64 | 200 | 47280 | 0.42 |
| Tissuemnist_28 | 200 | 47280 | 0.42 |
| tissuemnist_128 | 197 | 47280 | 0.41 |
| pathmnist_224 | 4 | 7180 | 0.055 |
| pathmnist_128 | 2 | 7180 | 0.027 |
| bloodmnist_128 | 0 | 3421 | 0 |
| bloodmnist_224 | 0 | 3421 | 0 |
| bloodmnist_64 | 0 | 3421 | 0 |
| Bloodmnist_28 | 0 | 3421 | 0 |
| pathmnist_64 | 0 | 7180 | 0 |
| Pathmnist_28 | 0 | 7180 | 0 |
| **Weighted Average** | | | 13.37 |
| **Macro Average** | | | 9.74 |

# F   Evaluation Details

## F.1   VQA evaluation details

Three out of four tasks are formulated as multiple-choice visual question answering: coarse-grained perception, fine-grained perception, and cognition. We describe their evaluation here, while the fourth object detection task is described separately in appendix G.1. For each sample, $i$, we have an image, $\mathbf{x}_i$ a question string, $q$, and a set of $k$ candidate answer strings $\{a_{ij}\}_{j=1}^k$.

For the autoregressive models, $f_A$, we first generate a query string, $t$, from the question and candidate answer strings, $t(q, \{a_{ij}\}_{j=1}^k)$, using the template in Figure 21. The prompt text instructs the response to start with a letter for one of the multiple choice answers, ('A', 'B', ... ). We pass the prompt string and image to the model and decode the output, $y = f_A(x_i, t)$, where $y$ is the output string. We have two strategies for matching the response string, $y$, to the candidate answers $\{a_{ij}\}_{j=1}^k$. First, for each $j$, we check whether the answer string is in the output: $a_{ij} \in y$ with lowercase string matching. This is important because models do not always follow the instructions to output the multiple choice letter and instead return the answer. If there were no matches, then we extract the first character from $y$. If it is one of ('A', 'B', ... ) – as instructed in the prompt – then that it is assigned to the corresponding answer, $a_{ij}$. Otherwise, we mark the answer as incorrect.

The contrastive models have a vision encoder $E_x$ and text caption encoder $E_c$. We first compute the image embedding, $z_{x_i} = E_x(\mathbf{x}_i)$. Then for each candidate answer, $a_{ij}$, we form a caption, $c$, as $c(a_{ij})$, using a text template that is suitable for CLIP-like models. The templates are in table 23 for the fine-grained perception task, and table 24 for the coarse-grained perception task. For the cognition task, each caption is the concatenation of the question string, $q$, and the candidate answer string, $a_{ij}$. We get the embedding for each caption, $z_{c_{ij}} = E_c(c_{ij})$ for $j \in [1, k]$, and then compute the cosine similarity score for each caption, $s_{ij} = z_{c_{ij}} \cdot z_{x_i}$ for $j \in [1, k]$. The $j$ with the largest $s_{ij}$ is the final prediction. If $\arg\max(s_{ij})$ has the same index as the correct answer the question is marked as correct, incorrect otherwise.

Code for evaluation is made public through our repository: eVLLM.

## F.2   Object detection evaluation details

We evaluate the two models that support localization: QwenVL [6] and PaliGemma [68], and follow their user guide for prompting. For 'QwenVL' the prompt is ``Detect {class_name}''. For PaliGemma the prompt is ``Detect {class_name}; {class_name}'', where the repeated class name indicates that multiple instances may be predicted. Our early experiments found that PaliGemma would sometimes fail to localize any instances using detection prompting, but would localize them with segmentation prompting. So if PaliGemma does return zero instances, we prompt for segmentation ``Segment {class_name}; {class_name}'' and extract the bounding box. The Burgess et al. dataset has two classes, so we prompt the model one at a time. Both models output detection predictions as a string with a standardized structure, which we parse using regex.

We use the GRIT localization metric [25] because it is well-motivated and has previously been used in VLM evaluations by QwenVL [6]. The score is:

$$\sum_{i=1}^{M} \frac{IoU_i}{P + G_{missed}} \tag{1}$$

There are $M$ ground truth boxes, and $P$ predicted boxes, which are matched using the Hungarian algorithm on the IoU metric. $G_{missed}$ is the number of predicted boxes not matched to a ground truth box. Intuitively, this metric measures the average IoU for matched boxes, while penalizing making too many predictions using $G_{missed}$ (similar to the precision metric). Note that we cannot use the more typical mAP score from object detection because they depend on a threshold for controlling the false-positive rate, which these VLMs do not support.

Table 18: Object detection results for all datasets with detection annotations and for all models that support object detection.

| Dataset | Class | PaliGemma | QwenVLM |
|---|---|---|---|
| **Easy** | | | |
| PCST-Contour | cell | 76.5 | 82.1 |
| PCST-Eccentricity | cell | 76.6 | 84.7 |
| PCST-Texture | cell | 78.0 | 85.9 |
| CellCognition (Golgi) | nucleus | 72.4 | 46.7 |
| CellCognition (H2B) | nucleus | 80.4 | 68.1 |
| CellCognition (Mt) | nucleus | 72.6 | 51.6 |
| | | | |
| **Hard** | | | |
| PCST-Contour | nucleus | 31.7 | 6.6 |
| PCST-Eccentricity | nucleus | 30.9 | 6.2 |
| PCST-Texture | nucleus | 30.2 | 6.4 |
| OpenCell | nucleus | 0.0 | 0.2 |
| Wu et al 2023 | mitochondria | 22.8 | 30.0 |
| GlaS Challenge | gland | 2.7 | 5.8 |

# G  Additional Benchmarking Results

## G.1  Object Detection Results

table 18 summarizes object detection for the datasets having object localization annotations.

Overall, the localization scores are very poor, which is expected since both models are generalist and object localization has received relatively less attention in autoregressive VLMs. Looking at the splits:

- **Easy split.** Although both models have higher scores for the 'cell' class in Burgess et al., they still fall below 80, and the task is straightforward for humans. The story is similar for 'nucleus' in Held et al., but for QwenVLM, the scores are even lower.

- **Hard split.** All models perform poorly on the hard split. In Burgess et al 'nucleus', PaliGemma scores around 30, however qualitative inspection shows that in most cases, the bounding box predicts the entire cell, which encapsulates the nucleus. Similarly, in Wu et al., the mitochondria class scores more than 20 for both models, but qualitative inspection shows that the prediction is usually a box around the entire image. We find the same pattern in 'Sirinukunwattana et al.' for 'gland' detection.

## G.2  Weight ensembling details

In the results section 5.2, we consider PLIP, which was fine-tuned from OpenCLIP, and QuiltNet, which was fine-tuned from CLIP, both using pathology data. Since we have benchmark results for all these models, our evaluation can evaluate the impact of fine-tuning on pathology data (moving from OpenCLIP/CLIP to PLIP/QuiltNet). We showed that pathology fine-tuning can improve performance on the pathology subset of Micro-Bench (the fine-grained perception), where we filter for all samples from the histopathology or H&E imaging modality. However, the fine-tuned models have worse overall performance on Micro-Bench(which includes the pathology subset).

We proposed to create 'merged models' by combining the base model (OpenCLIP or CLIP) with their fine-tuned models (PLIP or QuiltNet) with weight merging. Specifically, following [75], for a base model with weights $\theta_B$ and tuned model $\theta_T$ (which have the same architecture), the merged model weights $\theta_M$ are :

$$\theta_M = \alpha \cdot \theta_B + (1 - \alpha) \cdot \theta_T$$

That is, we are linearly interpolating each model weight independently, with a single fixed constant $\alpha$. We arbitrarily set $\alpha = 0.5$ in our experiments, but tuning that constant could lead to better overall results.

We now show more comprehensive results in table 1, which is the main results table but includes our merged models, M-PLIP and M-QuiltNet; table 20 is the same but with the pathology-only split. In all cases (M-PLIP and M-QuiltNet), the merged models outperform the tuned models (PLIP and QuiltNet), while the base models (OpenCLIP and CLIP) outperform almost all cases. For fine-grained perception (the task that is most in distribution for PLIP and QuiltNet training), the merged models become among the strongest performing overall models, and on pathology-specific fine-grained perception, the merged models outperform BiomedCLIP.

Table 19: Macro-average accuracy (with bootstrap confidence interval) for coarse-grained and fine-grained perception and cognition (reasoning) in Micro-Bench . Robust models (byproduct of merging fine-tuned models with their respective base models) are also included.

| $\mu$-Bench | | | | | |
|---|---|---|---|---|---|
| Perception (Coarse-Grained) | | Perception (Fine-Grained) | | Cognition (Reasoning) | |
| Model | Accuracy ($\pm$ CI) | Model | Accuracy ($\pm$ CI) | Model | Accuracy ($\pm$ CI) |
| GPT-4o | 62.68 ($\pm$ 0.35) | GPT-4o | 51.73 ($\pm$ 0.82) | GPT-4o | 62.00 ($\pm$ 9.00) |
| CogVLM | 52.05 ($\pm$ 0.35) | BiomedCLIP | 34.65 ($\pm$ 0.75) | QwenVLM | 41.00 ($\pm$ 10.00) |
| QwenVLM | 49.85 ($\pm$ 0.35) | CONCH | 33.64 ($\pm$ 0.72) | CogVLM | 41.00 ($\pm$ 10.00) |
| BiomedCLIP | 47.57 ($\pm$ 0.34) | M-PLIP* | 32.99 ($\pm$ 0.73) | OpenCLIP | 38.33 ($\pm$ 8.33) |
| M-PLIP* | 43.25 ($\pm$ 0.34) | M-QuiltNet* | 32.42 ($\pm$ 0.71) | M-PLIP* | 34.17 ($\pm$ 8.33) |
| ALIGN | 40.7 ($\pm$ 0.34) | ALIGN | 31.9 ($\pm$ 0.72) | ALIGN | 31.00 ($\pm$ 9.00) |
| OpenCLIP | 36.34 ($\pm$ 0.33) | CLIP | 30.09 ($\pm$ 0.71) | CLIP | 28.00 ($\pm$ 9.00) |
| PaliGemma | 36.29 ($\pm$ 0.33) | OpenCLIP | 29.36 ($\pm$ 0.69) | M-QuiltNet* | 25.83 ($\pm$ 7.52) |
| CLIP | 35.41 ($\pm$ 0.34) | CogVLM | 28.18 ($\pm$ 0.70) | BiomedCLIP | 25.00 ($\pm$ 8.00) |
| M-QuiltNet* | 31.26 ($\pm$ 0.32) | QuiltNet | 27.85 ($\pm$ 0.69) | PaliGemma | 25.00 ($\pm$ 8.00) |
| PLIP | 31.11 ($\pm$ 0.32) | QwenVLM | 27.81 ($\pm$ 0.70) | CONCH | 18.00 ($\pm$ 7.00) |
| CONCH | 27.84 ($\pm$ 0.31) | PLIP | 25.49 ($\pm$ 0.68) | Random | 17.00 ($\pm$ 7.00) |
| QuiltNet | 26.58 ($\pm$ 0.31) | PaliGemma | 21.29 ($\pm$ 0.64) | PLIP | 17.00 ($\pm$ 7.00) |
| Random | 18.34 ($\pm$ 0.27) | Random | 19.13 ($\pm$ 0.60) | QuiltNet | 13.00 ($\pm$ 6.00) |

\* General autoregressive VLMs ▪ General contrastive VLMS ▪ Pathology contrastive VLMS
▪ Biomedical contrastive VLMS.

### G.3 Model Performance on Pathology Specific Tasks

While prior evaluations show that general contrastive VLMs have some biology and pathology knowledge (a finding also reported in [31] and [34]), most specialist models analyzed in this work were fine-tuned. To this end, we analyzed the performance on pathology-only tasks and found similar rankings.

## H Additional Benchmarking details

### H.1 Model Details

Table table 21 provides a breakdown of model parameters and training data (with dataset size), specialist models include their base model.

### H.2 Computing confidence intervals

Error bars represent 95% confidence intervals (CI) computed via nonparametric bootstrapping using the SciPy $stats.bootstrap$ function with 1000 resamplings and default settings. No data were excluded from the analyses.

### H.3 Zero-shots results broken down by task

Figure 12 presents a breakdown of perception coarse-grained results by task. It reveals that autoregressive generalist models perform well across all tasks, as indicated by the overall averages. Notably, while GPT-4o dominates most tasks, PaliGemma excels in domain identification, achieving the best performance in that specific area.

Table 20: Macro-average accuracy (with bootstrap confidence interval) for coarse-grained and fine-grained perception (pathology only tasks) and cognition (reasoning) in Micro-Bench . Robust models (a byproduct of merging fine-tuned models with their respective base models) are also included.

| μ-Bench | | | | | |
|---|---|---|---|---|---|
| Perception (Coarse-Grained) | | Perception (Fine-Grained) | | Cognition (Reasoning) | |
| Model | Accuracy (± CI) | Model | Accuracy (± CI) | Model | Accuracy (± CI) |
| GPT-4o | 71.29 (± 0.45) | GPT-4o | 61.88 (± 1.13) | GPT-4o | 62.00 (± 9.00) |
| QwenVLM | 67.89 (± 0.47) | CONCH | 42.44 (± 1.13) | QwenVLM | 41.00 (± 10.00) |
| CogVLM | 59 (± 0.51) | M-PLIP* | 39 (± 1.12) | CogVLM | 41.00 (± 10.00) |
| ALIGN | 46.32 (± 0.50) | M-QuiltNet* | 36.95 (± 1.22) | OpenCLIP | 38.33 (± 8.33) |
| BiomedCLIP | 45.5 (± 0.51) | BiomedCLIP | 35.29 (± 1.09) | M-PLIP* | 34.17 (± 8.33) |
| PaliGemma | 43.34 (± 0.51) | QuiltNet | 33.28 (± 1.08) | ALIGN | 31.00 (± 9.00) |
| M-PLIP* | 42.27 (± 0.51) | OpenCLIP | 32.35 (± 1.09) | CLIP | 28.00 (± 9.00) |
| M-QuiltNet* | 37.68 (± 0.50) | PLIP | 32.02 (± 1.06) | M-QuiltNet* | 25.83 (± 7.52) |
| OpenCLIP | 37.59 (± 0.50) | CLIP | 28.63 (± 1.01) | BiomedCLIP | 25.00 (± 8.00) |
| CLIP | 31.92 (± 0.47) | ALIGN | 27.51 (± 0.99) | PaliGemma | 25.00 (± 8.00) |
| CONCH | 31.11 (± 0.32) | QwenVLM | 24.62 (± 0.97) | CONCH | 18.00 (± 7.00) |
| QuiltNet | 29.18 (± 0.45) | CogVLM | 23.8 (± 0.98) | Random | 17.00 (± 7.00) |
| PLIP | 22.72 (± 0.42) | PaliGemma | 22.77 (± 0.97) | PLIP | 17.00 (± 7.00) |
| Random | 18.15 (± 0.39) | Random | 17.74 (± 0.86) | QuiltNet | 13.00 (± 6.00) |

⁺ ▢ General autoregressive VLMs ▢ General contrastive VLMS ▢ Pathology contrastive VLMS ▢ Biomedical contrastive VLMS.

Figure 13 shows a breakdown of biology-specific perception fine-grained results by task. While GPT-4o dominates in some tasks, other models excel in specific tasks. For instance, ALIGN outperforms all models in molecular colocalization, while BiomedCLIP performs best in mitochondrial morphology classification.

Figure 14 shows a breakdown of pathology-specific perception fine-grained results by task. This breakdown reveals that while GPT-4o performs best in three tasks, specialist models still outperform it in amyloid morphology [a] and Pap smear grading.

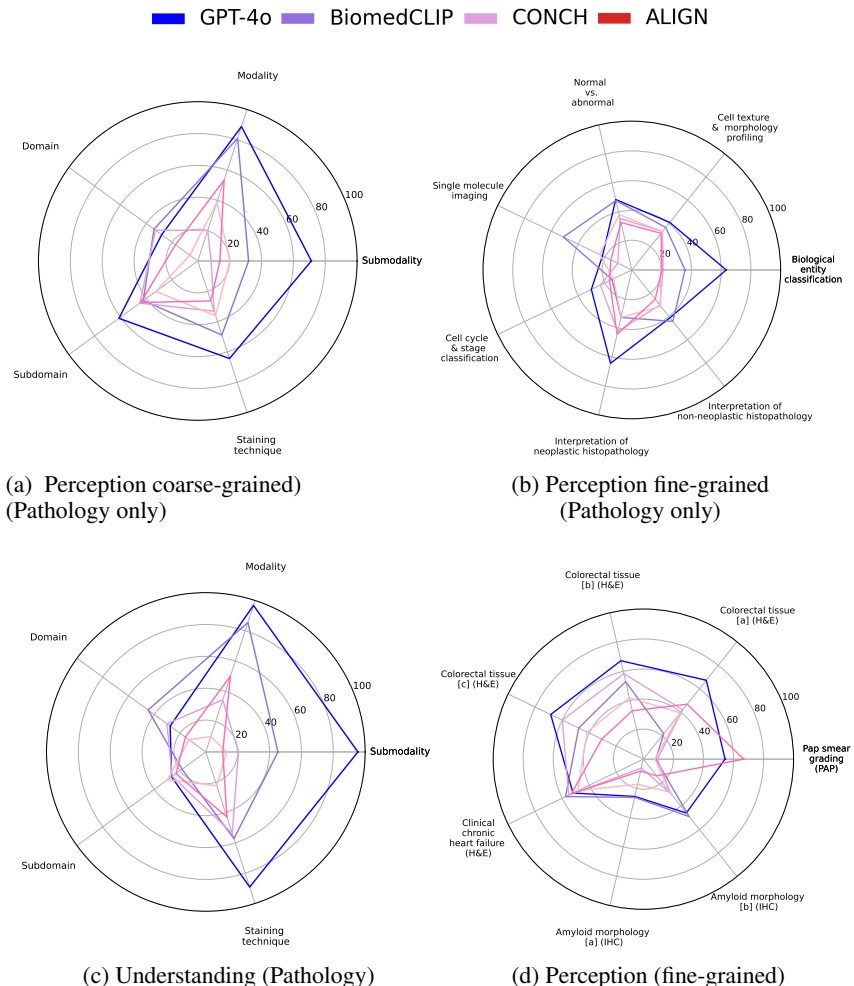

(a) Perception coarse-grained)
(Pathology only)

(b) Perception fine-grained
(Pathology only)

(c) Understanding (Pathology)

(d) Perception (fine-grained)

Figure 11: Performance comparison of top general domain auto-regressive models, contrastive models, and biomedical contrastive models across all axis of the Micro-Bench [perception]

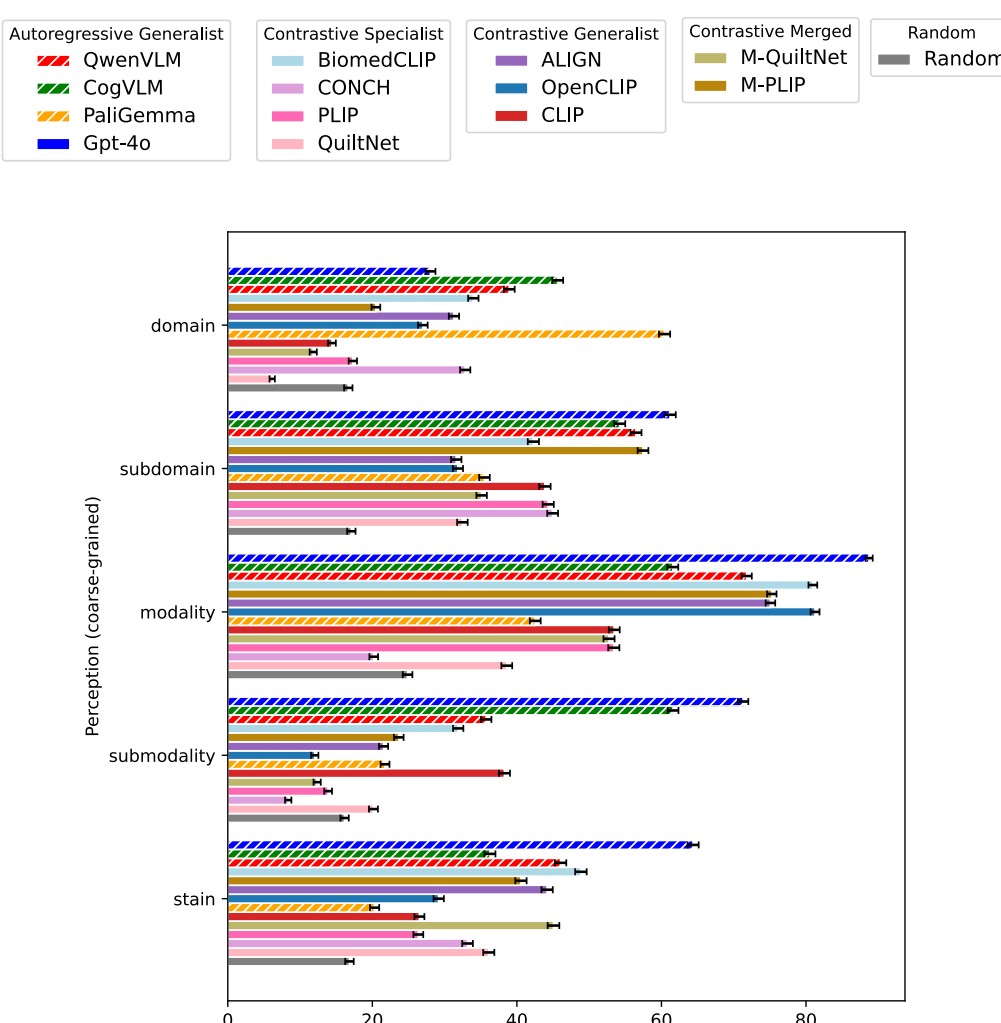

Figure 12: Perception (coarse-grained) results broken down by task

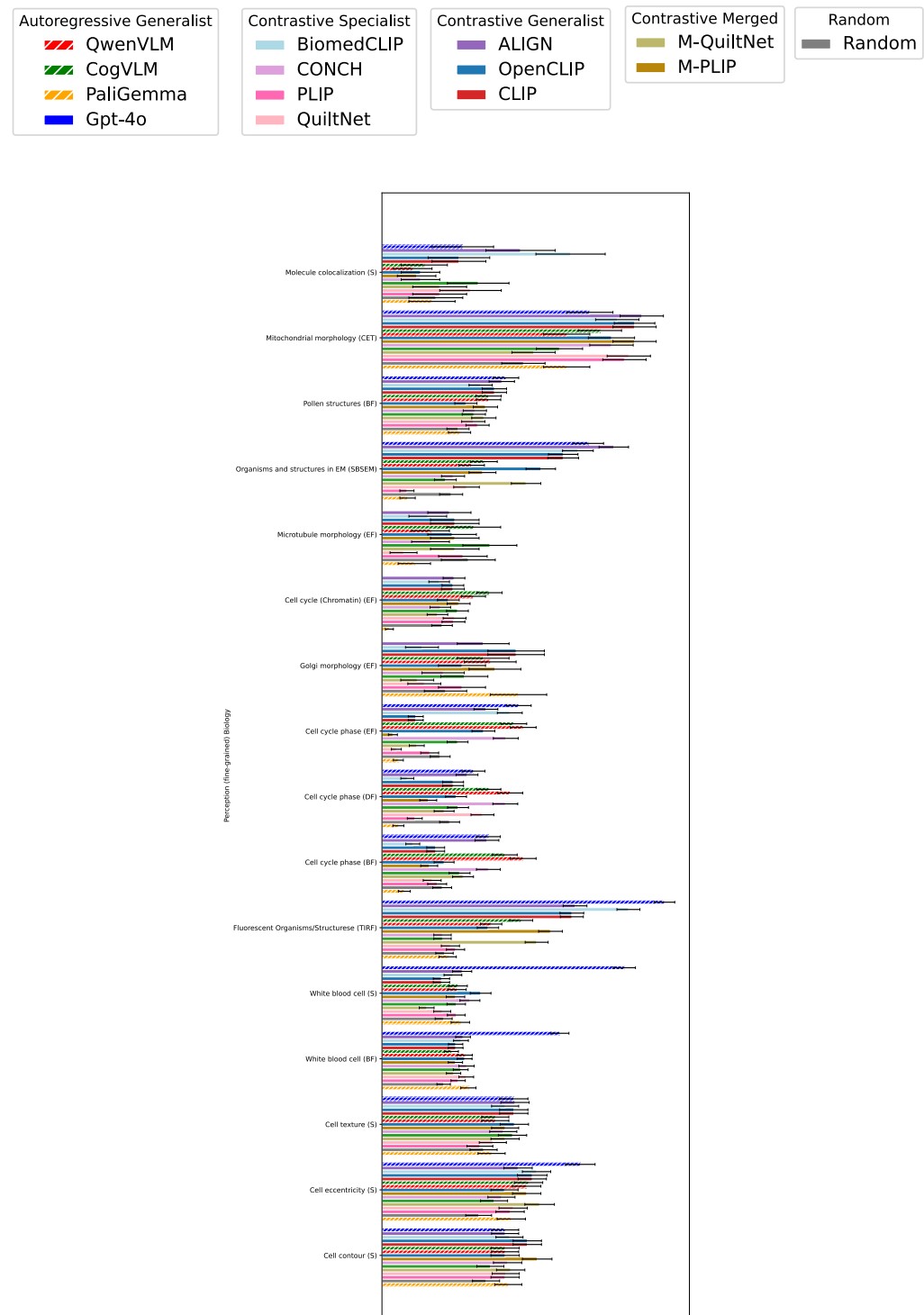

Figure 13: Perception (fine-grained) results broken down by task in biology

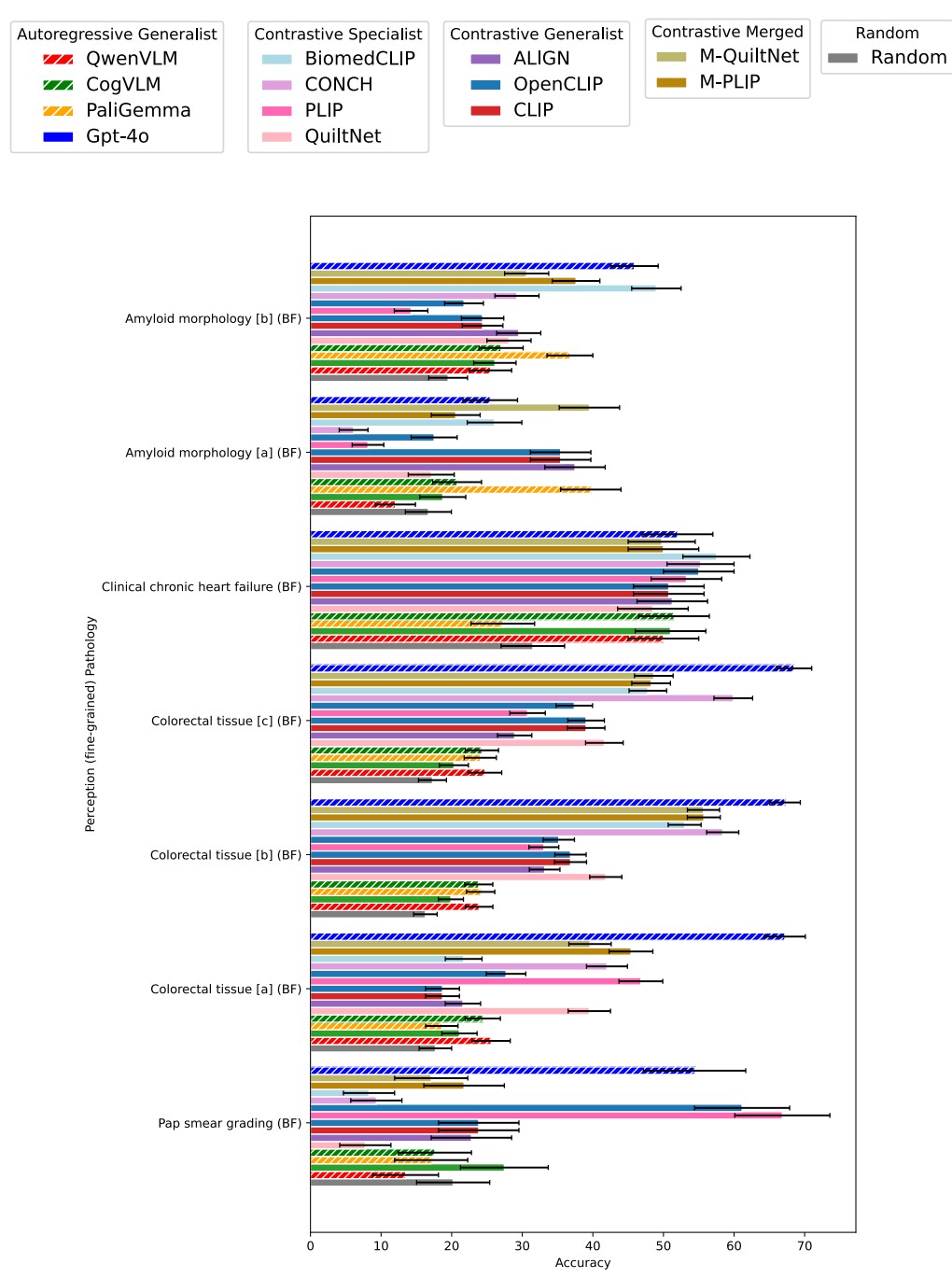

Figure 14: Perception (fine-grained) results broken down by task in pathology

Table 21: VLM Model Breakdown:

| Model | Total Params | Base VLM | Vision Encoder | VE Params | Text Encoder | TE Params | Training Data | Size |
|---|---|---|---|---|---|---|---|---|
| **Contrastive VLMs** | | | | | | | | |
| CLIP | 151.2M | - | ViT-B/32 | 86M | | | DataComp-1B | 13B |
| ALIGN [36] | 172.1M | - | EfficientNet | 62.1M | BERTbase | 109.4M | Internet | 1.8B |
| CoCa [83] | 383M | - | ViT-B/16 | 86M | | 297M | Internet | 4.8B |
| OpenCLIP | 223.7M | - | ViT-B/32 | 86M | | | DataComp-1B | 13B |
| BLIP [45] | 223.7M | - | ViT-B/16 | 86M | BERTbase | 137.2M | Internet | 14M |
| | | | | | | | | |
| PLIP [32] | 151.2M | CLIP* | ViT-B/32 | 86M | | | OpenPath (X) | 208.4K |
| QuiltNet [34] | 151.2M | CLIP* | ViT-B/16 | 86M | GPT2 (77CL) | | Quilt | 1M |
| BiomedCLIP | 195.M | OpenCLIP | ViT-B/16 | 86M | BioMedBERT [13] | 110M | PMC-15M | 15M |
| CONCH [49] | 395.2M | CoCa | ViT-B/16 | 86M | | 1.17M | | |
| | | | | | | | | |
| **Auto-regressive VLM** | | | | | | | | |
| CogVLM [71] | 17.6B | - | EVA2-CLIP-E | | Vicuna-1.5-7B | 7B | Multiple | 1.5B |
| Qwen-VL [6] | 9.6B | - | ViT-bigG | 1.9B | QwenLM | 7.7B | Multiple | 1.4B |

# I   VQA Templates

Table 22: Micro-Bench Perception Coarse-Grained Question and Caption templates

| Type | Question | Caption |
|---|---|---|
| **Modality** | What is the most likely microscopy modality used to acquire this image? | A microscopy image obtained through {modality}. |
| **Submodality** | What is the most likely microscopy submodality used to acquire this image? | A microscopy image obtained through {submodality}. |
| **Domain** | What is the most likely field of study this micrograph would be used for? | A microscopy image frequently studied in {domain}. |
| **Subdomain** | What is the most likely subfield of study this micrograph would be used for? | A microscopy image frequently studied in {subdomain}. |
| **Stain** | What is the most likely technique used to stain this micrograph? | A microscopy image stained with {stain}. |

Table 23: Micro-Bench Perception Fine-grained: Question and caption templates for biology tasks

| Dataset | Question | Caption |
|---|---|---|
| PCST-Contour | A synthetic fluorescence micrograph is displayed. What is the most likely description for the cells contour irregularities? | A synthetic fluorescence microscopy image of a cell with {class} contours. |
| PCST-Eccentricity | A synthetic fluorescence micrograph displayed. What is the most likely description for the cells eccentricity phenotype? | A synthetic fluorescence microscopy image of a cell with {class} eccentricity. |
| PCST-Texture | A synthetic fluorescence micrograph displayed. What is the most likely description for the cells cytoplasm texture? | A synthetic fluorescence microscopy image of the cell cytoplasm with {class} texture. |
| Colocalization benchmark | A synthetic confocal fluorescence micrograph of small points in two different channels with different levels of colocalization. Given the image provided, what is the most accurate description for the colocalization patterns? | A synthetic confocal fluorescence microscopy image of small points in two different channels with different levels of colocalization. The image displays {class} colocalization patterns. |
| EMPIAR SBF-SEM | An electron micrograph is shown. Based on the image, what is the most likely structure on the field of view? | A Serial blockface scanning electron microscopy image shows {class} |
| BBBC048 (Brightfield) | A brightfield micrograph of jurkat cells acquired using flow cytometry (single cell). Based on the micrograph, what is the most likely cell phase? | Brightfield microscopy imaging flow cytometry is used to visualize single Jurkat cells at different cell cycle phases. The image displays a cell in {class} stage of the cell cycle. |
| BBBC048 (Darkfield) | A darkfield micrograph of jurkat cells acquired using flow cytometry (single cell). Based on the micrograph, what is the most likely cell phase? | A darkfield microscopy imaging flow cytometry is used to visualize single jurkat cells at different cell cycle phases. The image displays a cell in {class}. |
| BBBC048 (Epifluorescence) | A propidium iodide stained fluorescence micrograph of Jurkat cells acquired using flow cytometry (single cell). Based on the micrograph, what is the most likely cell phase? | Epifluorescence microscopy imaging (flow cytometry) shows single Jurkat cells stained with propidium iodide at different cell cycle phases. The image displays a cell in {class}. |
| CellCognition (Golgi) | A fluorescence micrograph of Hela Kyoto cells stably expressing GalT-eGFP to label the Golgi apparatus. Based on the image what is the most likely the Golgi apparatus morphology? | Fluorescence microscopy image of human Hela Kyoto cells stably expressing galactosyltransferase (GalT-eGFP) to label the Golgi apparatus showing {class} morphology. |
| CellCognition (H2B) | A fluorescence microscopy image of human Hela Kyoto cells with stable chromatin marker expression. Based on the image, what is the most likely cell cycle stage? | Fluorescence microscopy image of human Hela Kyoto cells with stable chromatin marker expression. The micrograph displays a cell in {class} stage of the cell cycle. |
| CellCognition (Mt) | A fluorescence micrograph of Hela Kyoto cells stably expressing eGFP-labeled tubulin to label microtubules is shown. Based on the micrograph what is the most likely microtubule morphology? | Fluorescence microscopy image of human Hela Kyoto cells with stable chromatin marker expression (eGFP) displays microtubules showing {class} morphology. |
| ICPR2020 Pollen | Basic fuchsin stained light micrograph of pollen grains. Based on the image, what is the most likely pollen class? | A brightfield microscopy image of pollen grains shows {class} structures. |
| Wu et al 2023 | A cryo-electron tomography of mitochondria in neurons cultured in vitro is shown. Based on the image what is the most likely mitochondrial morphology? | A cryo-electron tomography of mitochondria in neurons cultured in vitro shows {class}. |
| Fluorescence Cells & Structures | A fluorescence micrograph is shown. Based on the image, what is the most likely structure? | A photomicrograph shows a fluorescence microscopy {class}. |

Table 24: Micro-Bench Perception Fine-grained: Question and caption templates for pathology tasks

| Dataset | Question | Caption |
|---|---|---|
| Acevedo et al 2020 | A Giemsa-stained light micrograph displaying human peripheral blood cells. As a blood cell recognition system, identify the correct cell type: | A brightfield microscopy image of a peripheral blood smear stained with giemsa displaying {article} {class}. |
| Jung et al 2022 | Synthetically generated Giemsa-stained light micrograph of human peripheral blood cell. As a blood cell recognition system, identify the correct cell type: | A synthetic microscopy image of a peripheral blood smear stained with giemsa, displays {article} {class}. |
| Kather et al 2016 | H&E stained light micrograph of human colorectal tissue. Based on the image, what is the most likely texture class? | H&E stained light microscopy image of human colorectal tissue with {class}. |
| Kather et al 2018 | H&E stained light micrograph of human colorectal tissue. Based on the image, what is the most likely texture class? | H&E stained light microscopy image of human colorectal tissue with {class}. |
| Kather et al 2018 Val7K | H&E stained light micrograph of human colorectal tissue. Based on the image, what is the most likely texture class? | H&E stained light microscopy image of human colorectal tissue with {class}. |
| Nirschl et al 2018 | H&E stained light micrograph of human cardiac tissue. Based on the image, what is the most likely clinical chronic heart diagnosis? | H&E stained light microscopy image of human cardiac tissue with {class} texture. |
| Tang et al 2019 | IHC stained light micrograph of extracellular amyloid-beta deposition in the human brain tissue. Based on the image, what is the most likely amyloid beta morphology pattern? | Human brain tissue is stained with immunohistochemistry for amyloid-beta and imaged using brightfield microscopy. The micrograph displays {class} morphology. |
| Wong et al 2022 | IHC stained light micrograph of extracellular amyloid-beta deposition in the human brain tissue. Based on the image, what is the most likely amyloid beta morphology pattern? | Human brain tissue is stained with immunohistochemistry for amyloid-beta and imaged using brightfield microscopy. The micrograph displays {class} morphology. |

# J Taxonomy

```
light microscopy:
    - brightfield microscopy
    - phase contrast microscopy
    - differential interference contrast microscopy
    - darkfield microscopy
    - polarized light microscopy
    - mixed
    - synthetic
fluorescence microscopy:
    - confocal microscopy
    - epifluorescence microscopy
    - single-molecule localization microscopy (SMLM)
    - stimulated emission depletion microscopy (STED)
    - total internal reflection fluorescence microscopy (TIRF)
    - fluorescence recovery after photobleaching (FRAP)
    - fluorescence resonance energy transfer (FRET)
    - fluorescence in situ hybridization (FISH)
    - fluorescence correlation spectroscopy (FCS)
    - mixed
    - synthetic
electron microscopy:
    - atomic force microscopy
    - scanning electron microscopy
    - serial blockface scanning electron microscopy
    - cryo-electron microscopy
    - cryo-electron tomography
    - immuno-electron microscopy
    - mixed
    - synthetic
```

Figure 15: Modality with submodality YAML file.

```yaml
biology:
    - anatomy
    - biochemistry
    - biophysics
    - biotechnology
    - botany
    - cell and molecular biology
    - cell biology
    - cell cycle
    - conservation biology
    - developmental biology
    - ecology
    - evolutionary biology
    - genetics
    - immunology
    - marine biology
    - microbiology
    - neurobiology
    - parasitology
    - pharmacology
    - physiology
    - structural biology
    - systems biology
    - virology
    - zoology
dermatology:
    - infectious dermatology
    - medical dermatology
    - neoplastic dermatology
ophthalmology:
    - cornea and external eye ophthalmology
    - retinal surgery
    - diabetic retinopathy
    - neuro-ophthalmology
    - pediatric ophthalmology
    - neoplastic ophthalmology
    - medical ophthalmology
cytology:
    - gynecologic cytology
    - non-gynecologic cytology
    - fine needle aspiration cytology
pathology:
    - autopsy pathology
    - blood banking and transfusion medicine
    - bone and soft tissue pathology
    - breast pathology
    - cardiovascular pathology
    - clinical pathology
    - dermatopathology
    - endocrine pathology
    - forensic pathology
    - gastrointestinal pathology
    - genitourinary pathology
    - gynecologic pathology
    - head and neck pathology
    - hematopathology
    - hepatobiliary pathology
    - infectious disease pathology
    - molecular pathology
    - nephropathology
    - neuropathology
    - oral pathology
    - ophthalmic pathology
    - pancreatic pathology
    - pediatric pathology
    - pulmonary and pleural pathology
    - renal and medical kidney pathology
    - surgical pathology
radiology:
    - abdominal radiology
    - breast imaging
    - cardiothoracic radiology
    - emergency radiology
    - gastrointestinal radiology
    - genitourinary radiology
    - head and neck radiology
    - interventional radiology
    - musculoskeletal radiology
    - neuroradiology
    - nuclear radiology
    - pediatric radiology
    - vascular and interventional radiology
```

Figure 16: Domain with subdomain YAML file.

```yaml
light microscopy:
    - H&E
    - IHC(HDab)
    - IHC(Red)
    - Giemsa
    - PAS
    - Papanicolaou
    - Masson Trichrome
    - Toluidine Blue
    - Wright-Giemsa
    - Ziehl-Neelsen
    - Gram
    - Congo Red
    - Alcian Blue
    - Basic fuchsin
    - None
    - synthetic
fluorescence microscopy:
    - DAPI
    - Hoechst
    - propidium iodide
    - SYTOX
    - Alexa Fluor 350 # blue
    - Alexa Fluor 405
    - GFP
    - FITC
    - Cy2
    - Alexa Fluor 488
    - RFP
    - Cy3
    - H2B-mCherry
    - Texas Red
    - Alexa Fluor 555
    - Alexa Fluor 568
    - Cy5
    - Alexa Fluor 647
    - Alexa Fluor 660
    - AlexaFluor-tubulin
    - synthetic
    - GalT-EGFP d
electron microscopy:
    - uranyl acetate
    - osmium tetroxide
    - lead citrate
    - phosphotungstic acid
    - tannic acid
    - sodium silicotungstate
    - sodium phosphotungstate
    - sodium metaperiodate
    - synthetic
    - None
```

Figure 17: Modality with submodality YAML file.

```
I'm creating a dataset to evaluate VLM understanding on biomedical images.
Could you convert this user input question into a multi-choice question
with 6 answer choices? One choice should be "None of the above",
and this choice should have a 1/6 chance of being correct.
Output a JSON format:
{"question": str, "choices": list, "answer": int (start from 0)}.

Context: {question["CONTEXT"]}
Input Question: {question["INPUT"]}
Correct Answer: {answer}
```

Figure 18: Prompt used to convert open VQA to closed VQA

```
Given this question, can you help me annotate the following fields?

## Modality and Submodality
{Modalities YAML}

## Domain and Subdomain
{Domains YAML}

## Scale (nano/subcellular, micro/cellular, macro/tissues)
{Scales Table}

## Content
+ Gene pathways
+ Metabolic pathways
+ Cell signalling and signal transduction
+ Cell physiology/function
+ Protein-protein interactions
+ Cell-cell interaction
+ Unique properties of the cell of origin/cell type in the image
+ Cytoskeleton and cell structure/morphology
+ Drug or small molecule mechanism of action
+ Other

## Relevant biological keywords
For example, brain, HeLa, mitochondria, GFP, etc

Output a JSON with {"modality": str,
                     "submodality": str,
                     "domain": str,
                     "subdomain": str,
                     "scale": str,
                     "content": str,
                     "keywords": list[str]}
```

Figure 19: Prompt use to classify questions post-hoc

```
metadata:
  height: 250
  width: 250
  name: 01145_6de79663_33375_split-test_chronic-heart-failure.png
  format: .png
  createdAt: '2024-05-27T17:39:08.866Z'
  updatedAt: '2024-05-27T17:39:08.866Z'
comments: []
custom_metadata:
  age: 47.0
  classes_to_idx:
    not_chronic_heart_failure: 0
    chronic_heart_failure: 1
  cvdo_id:
  - CVDO_0000569
  cvdo_name:
  - cardiomyopathy
  dataset_name: nirschl_et_al_2018
  dataset_slug: nirschl_et_al_2018
  domain: pathology
  ethnicity: Caucasian
  filename: 01145_6de79663_33375_split-test_chronic-heart-failure.png
  file_size: 141080
  institution:
  - upenn
  image_id: 6de79663-0392-4f20-b2ea-16ddf4e3b4e4
  image_md5: 97ee369881fcbfd18fb4fb8dcfe2ca17
  label: 1
  label_name: chronic_heart_failure
  label_subname: cardiomyopathy
  label_task: classification of heart failure using cardiac histopathology images
  last_updated: '2024-05-27T17:39:08.868Z'
  license: CC-BY-4.0
  microns_per_pixel: 2.0
  modality: light microscopy
  ncbitaxon_id:
  - NCBITaxon_9606
  ncbitaxon_name:
  - Homo sapiens
  ncit_id:
  - NCIT_C50577
  normal_or_abnormal: abnormal
  original_filename: 33375_0_fal_20_0.png
  patient_id: '33375'
  pato_id:
  - PATO_0000384
  sex: male
  snomedct_id:
  - SCTID_48447003
  split: test
  stain: H&E
  subdomain: cardiovascular pathology
  submodality: brightfield microscopy
  supported_tasks:
  - multi_class
  uberon_id:
  - UBERON_0000948
  uberon_name:
  - heart
  questions: null
tags:
- CVDO_0000569
- H&E
- Homo sapiens
- NCBITaxon_9606
- NCIT_C50577
- PATO_0000384
- SCTID_48447003
- UBERON_0000948
- brightfield microscopy
- cardiomyopathy
- cardiovascular pathology
- heart
- light microscopy
- pathology
```

Figure 20: Micro-Bench Perception: Example of densely annotated metadata for a single data point. Metadata is collected and reviewed by an expert.

```
Answer with a single letter, no extra details.
Question: {question}
{options}
```

Figure 21: Prompts used to run inference with auto-regressive models

You are invited to an alpha-testing phase of a vision-language chat app
for biologists. The application is free of charge, poses no risks that
would not be present with general internet usage, and you may stop use
at any time. You may use it for your daily research or however you wish.

Website:
    Create an account at: ###
    Acknowledge terms of service
    Registration and use of the app is consent to use the submitted
    image/text for model training and testing purposes. The user's field
    of study and training level will be recorded. However, all personal
    information will remain confidential. The users will retain the
    copyright and ownership of the input image data. However, the terms
    of service allow permission to use and redistribute the image under
    a CC-BY-SA 4.0 license. The raw and/or curated image-text data may be
    used to create a public benchmark of real-world biology user-AI
    assistant instruction tuning dataset to benefit the biomedical
    computer vision community.

Main interface
1. Upload a biology or biomedical image.
    Describe the image as context for the model. For example:
    "Actin (orange) and mitochondria (cyan) in a micropatterned HeLa cell.
    This is a still image from time-lapse live cell imaging. Wild-type
    genotype and no drug treatment."

2. Prompt/question:
    For example: How would you describe the pattern of the
    organelle in cyan?
    What is the most likely organelle? What antibody marker or dye
    specifically labels this organelle for cell biology experiments?
    Feel free to challenge the model with difficult questions requiring
    complex reasoning. There is no need to limit questions to simple
    perceptual tasks such as classification ("what is in the image?")
    or visual question-answering. Have it reason and interpret
    images in a way that would challenge a new biology graduate student.
    Ask the model to do complex image-based reasoning about biological
    processes/pathways:

    "Given the gene knockout cell image provided, what biological pathway
    is most likely disrupted, if any?"

    "Are there any small molecules that reverse this phenotype?"

    "What diseases are associated with gene <my favorite gene>?"

    Determine whether the VLM can understand true biological signals vs.
    artifacts that confound analysis, See how the VLM handles diverse
    modalities and experiments (EM, fluorescence, brightfield, CLEM etc)
    as well as different cell types and tissues.
    Ask the model to generate new hypotheses based on an image or connect
    to relevant literature.

    After you have a response to your question, we encourage you to
    provide feedback on the VLM answer. Please give details on why the
    answer is correct/helpful or incorrect/not helpful. As needed,
    provide additional details as to whether the VLM
        1) understood the  question
        2) identified the biological feature(s) in the image
        3) provided a correct biological interpretation/answer.

52

Figure 22: Email Invitation: Invitation for collaboration send to submitters.

# K Micro-Bench Perception (Coarse-Grained) Closed VQA Data Samples

| QUESTION TYPE: Modality |
|---|

**Question:** What is the most likely microscopy modality used to acquire this image?
**Options:**
A. electron microscopy
B. fluorescence microscopy
C. light microscopy
D. none of the above

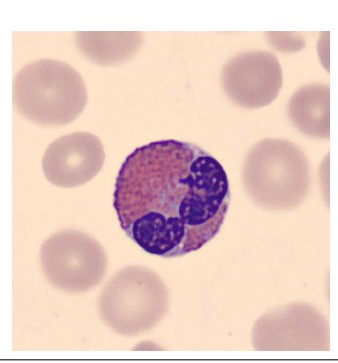

| QUESTION TYPE: Modality |
|---|

**Question:** What is the most likely microscopy modality used to acquire this image?
**Options:**
A. light microscopy
B. electron microscopy
C. fluorescence microscopy
D. none of the above

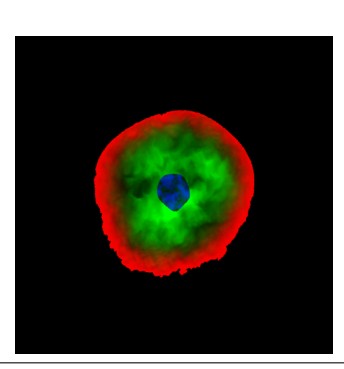

| QUESTION TYPE: Submodality |
|---|

**Question:** What is the most likely microscopy submodality used to acquire this image?
**Options:**
A. fluorescence resonance energy transfer (FRET)
B. total internal reflection fluorescence microscopy (TIRF)
C. mixed
D. stimulated emission depletion microscopy (STED)
E. confocal microscopy
F. none of the above

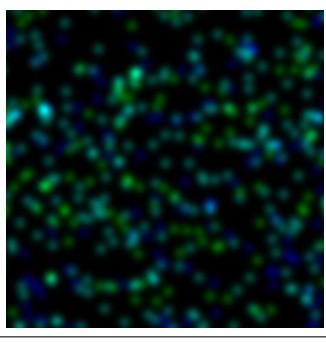

| QUESTION TYPE: Submodality |
|---|

**Question:** What is the most likely microscopy submodality used to acquire this image?
**Options:**
A. serial blockface scanning electron microscopy
B. atomic force microscopy
C. synthetic
D. cryo-electron microscopy
E. mixed
F. none of the above

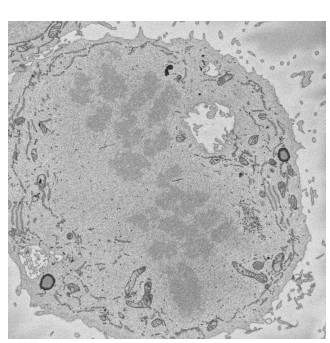

| QUESTION TYPE: Domain |
|---|

**Question:** What is the most likely field of study this micrograph would be used for?
**Options:**
A. ophthalmology
B. biology
C. radiology
D. pathology
E. cytology
F. none of the above

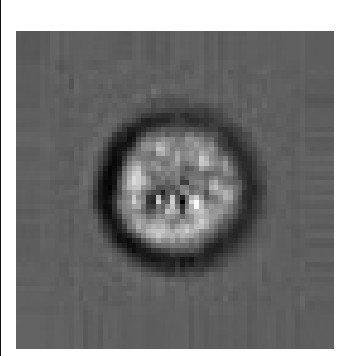

| QUESTION TYPE: Domain |
|---|

**Question:** What is the most likely field of study this micrograph would be used for?
**Options:**
A. cytology
B. radiology
C. ophthalmology
D. pathology
E. biology
F. none of the above

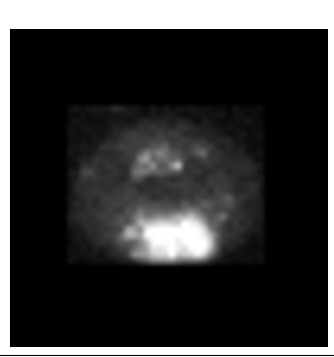

| QUESTION TYPE: Subdomain |
| --- |

**Question:** What is the most likely subfield of study this micrograph would be used for?
**Options:**
A. fine needle aspiration cytology
B. gynecologic cytology
C. non-gynecologic cytology
D. none of the above

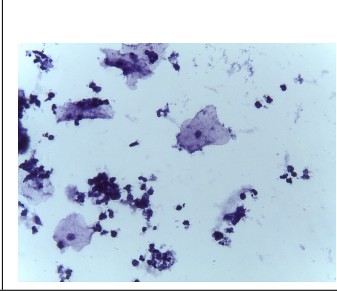

| QUESTION TYPE: Subdomain |
| --- |

**Question:** What is the most likely subfield of study this micrograph would be used for?
**Options:**
A. biotechnology.
B. botany.
C. neurobiology.
D. physiology.
E. biology.
F. none of the above

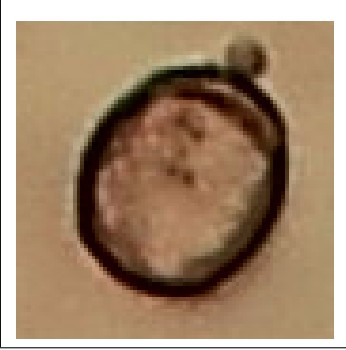

| QUESTION TYPE: Stain |
| --- |

**Question:** What is the most likely technique used to stain this micrograph?
**Options:**
A. Papanicolaou
B. H&E
C. PAS
D. IHC(HDab)
E. Giemsa
F. none of the above

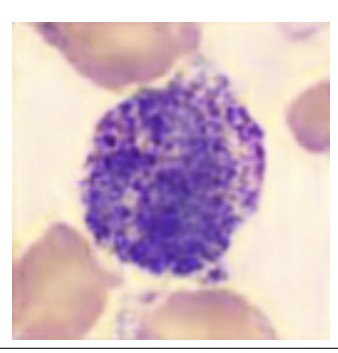

| QUESTION TYPE: Stain |
|---|

**Question:** What is the most likely technique used to stain this micrograph?
**Options:**
A. Giemsa
B. Ziehl-Neelsen
C. IHC(Red)
D. H&E
E. Alcian Blue
F. none of the above

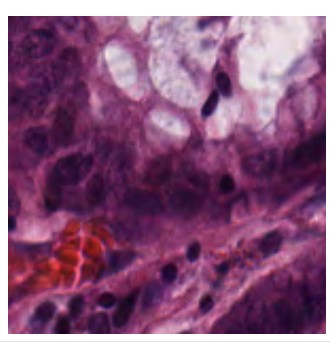

# L  Micro-Bench Perception (Fine-Grained) Closed VQA Data Samples

| TASK: White blood cell classification |
|---|

**Prompt:** A Giemsa-stained light micrograph displaying human peripheral blood cells. As a blood cell recognition system, identify the correct cell type:
**Options:**
A. eosinophil
B. neutrophil
C. immature granulocyte
D. platelet
E. erythroblast
F. none of the above

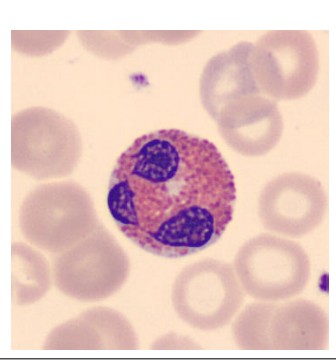

| TASK: Classification of cell contour irregularity phenotypes in synthetic images |
|---|

**Prompt:** A synthetic fluorescence micrograph is displayed. What is the most likely description for the cell's contour irregularities?
**Options:**
A. irregular
B. intermediate
C. smooth
D. none of the above

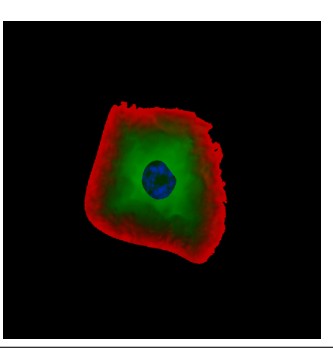

| TASK: Classification of cell contour irregularity phenotypes in synthetic images |
|---|

**Prompt:** An electron micrograph is shown. Based on the image, what is the most likely structure on the field of view?
**Options:**
A. a Zebrafish retina
B. Leishmania haptomonad
C. a HeLa cell in metaphase
D. Cardiac muscle
E. a Tobacco leaf chloroplast
F. none of the above

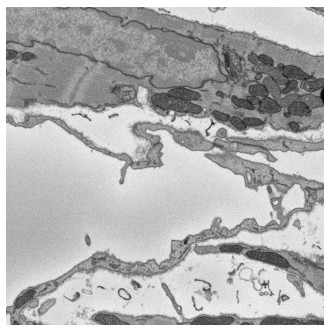

| TASK: Classification of cell cycle phase |
|---|

**Prompt:** A brightfield micrograph of Jurkat cells acquired using flow cytometry (single cell). Based on the micrograph, what is the most likely cell phase?
**Options:**
A. interphase (G2)
B. interphase (G1) phase
C. Telophase
D. Anaphase
E. Synthesis
F. none of the above

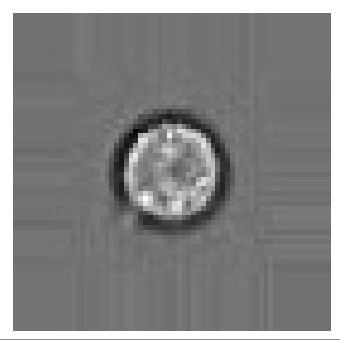

| TASK: Classification of cell cycle stages in live cell imaging data |
|---|

**Prompt:** A fluorescence micrograph of Hela Kyoto cells stably expressing GalT-eGFP to label the Golgi apparatus. Based on the image what is the most likely the Golgi apparatus morphology?
**Options:**
A. Anaphase
B. Diffuse
C. Interphase
D. Golgi twin
E. Partly disassembled
F. none of the above

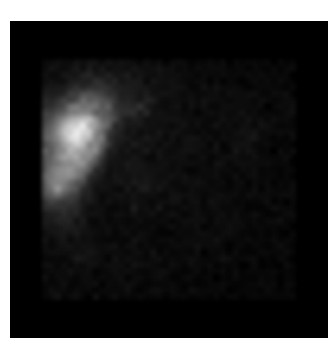

| TASK: Classification of pre-cancerous and cervical cancer lesions in liquid-based cytology Pap smear images |
|---|

**Prompt:** Liquid-based cytology pap smear of human pre-cancerous or cervical cancer lesions. Based on the cytogram, what is the most likely finding?
**Options:**
A. Squamous cell carcinoma (SCC)
B. Low-grade (LSIL)
C. Negative (NILM)
D. High-grade (HSIL)
E. none of the above

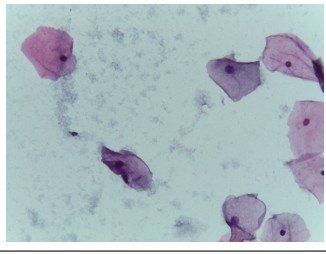

| TASK: Classification of normal and abnormal pollen grains |
|---|

**Prompt:** Basic fuchsin stained light micrograph of pollen grains. Based on the image, what is the most likely pollen class?
**Options:**
A. abnormal Corylus avellana
B. Non-pollen
C. normal Alnus
D. normal Corylus avellana
E. none of the above

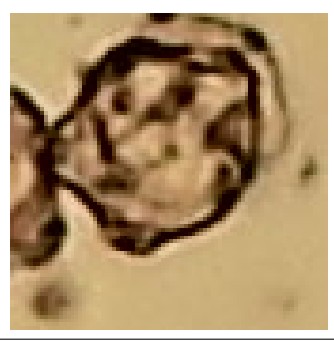

| **TASK: White blood cell classification** |
|---|

**Prompt:** Synthetically generated Giemsa-stained light micrograph of human peripheral blood cell. As a blood cell recognition system, identify the correct cell type:

**Options:**
A. Neutrophil
B. Basophil
C. Lymphocyte
D. Eosinophil
E. Monocyte
F. none of the above

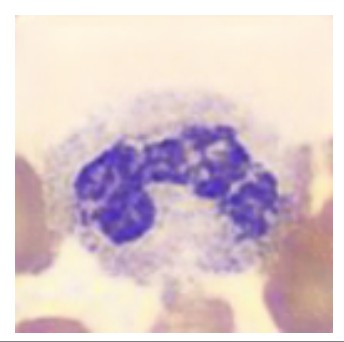

| **TASK: Texture classification in colorectal cancer** |
|---|

**Prompt:** H&E stained light micrograph of human colorectal tissue. Based on the image, what is the most likely texture class?

**Options:**
A. chronic heart failure
B. not chronic heart failure
C. none of the above

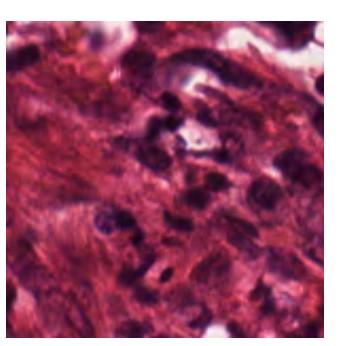

| **TASK: Classification of heart failure using cardiac histopathology images** |
|---|

**Prompt:** H&E stained light micrograph of human cardiac tissue. Based on the image, what is the most likely clinical chronic heart diagnosis?

**Options:**
A. chronic heart failure
B. not chronic heart failure
C. none of the above

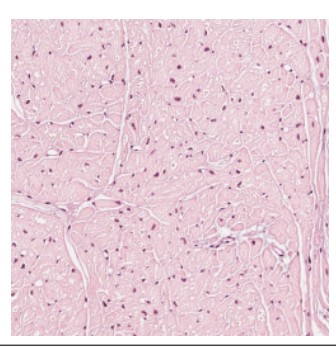

| **TASK: Amyloid beta pathology classification** |
| --- |

**Prompt:** IHC stained light micrograph of extracellular amyloid-beta deposition in the human brain tissue. Based on the image, what is the most likely amyloid beta morphology pattern?

**Options:**
A. Caa
B. Cored
C. Negative
D. Diffuse
E. none of the above

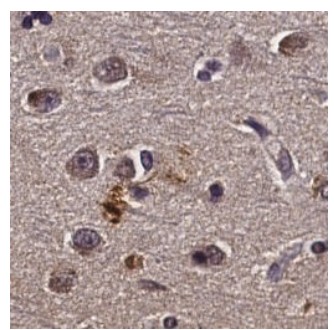

| **TASK: Classification of mitochondrial morphology in CryoET images** |
| --- |

**Prompt:** A cryo-electron tomography of mitochondria in neurons cultured in vitro is shown. Based on the image what is the most likely mitochondrial morphology?

**Options:**
A. abnormal mitochondrial morphology
B. normal mitochondrial morphology
C. none of the above

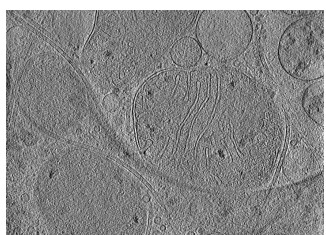

## M  Micro-Bench Cognition Closed VQA Data Samples

| Cognition Question Example |
|---|

**Question:** What type of cells are labelled green?
**Options:**
A. Cholinergic neurons
B. Glial cells
C. Muscle fibers
D. Epithelial cells
F. Fibroblasts
G. None of the above

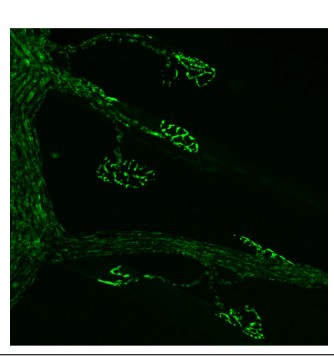

| Cognition Question Example |
|---|

**Question:** In the provided fluorescence microscopy image of a section of a human pancreas stained with DAPI (dark blue), anti-insulin (light blue), anti-glucagon (red), and anti-somatostatin (green), are there more green-labeled or red-labeled features visible?
**Options:**
A. There are more green-labeled features (anti-somatostatin).
B. There are more red-labeled features (anti-glucagon).
C. The number of green-labeled features (anti-somatostatin) and red-labeled features (anti-glucagon) are equal.
D. There are no green-labeled features (anti-somatostatin) visible.
E. There are no red-labeled features (anti-glucagon) visible.
F. None of the above

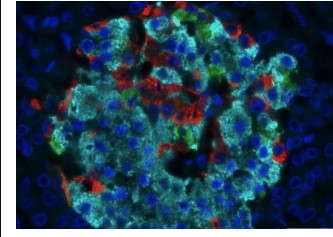

**Cognition Question Example**

**Question:** What are the dark structures in the image? How does it relate to the biological process demonstrated in the image?

**Options:**

A. The dark structures are mitochondria, involved in energy production during cell metabolism.

B. The dark structures are lysosomes, which digest cellular waste as the cell divides.

C. The dark structures are chloroplasts, which harvest light energy during photosynthesis in plant cells.

D. The dark structures are vesicles, which transport materials to the cleavage furrow during cytokinesis.

E. The dark structures are homologous chromosomes, separated into each daughter cell during mitosis.

F. None of the above

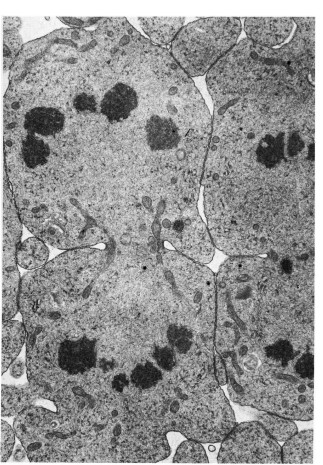

---

**Cognition Question Example**

**Question:** What is the most likely structure labeled by the bright dots along the neurons?

**Options:**

A. Synapse parts

B. N-cadherin complexes

C. V-glut 1 and 2 transporters

D. Postsynaptic NMDA receptors

E. Excitatory synapses

F. None of the above

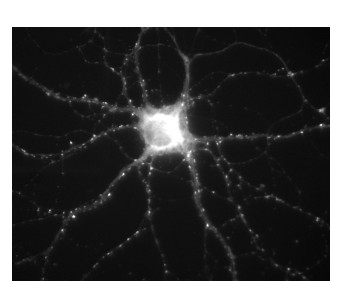

---

**Cognition Question Example**

**Question:** What type of imaging is this?

**Options:**

A. This is an image of HT55 cancer cells and T cells.

B. This is an image of neuronal cells.

C. This is an image of bacterial colonies.

D. This is an image of red blood cells.

E. This is an image of muscle tissue.

F. None of the above.

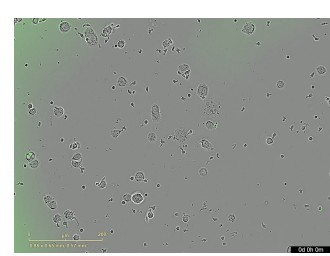

## Cognition Question Example

**Question:** What percentage of this tumor is composed of calcium?

**Options:**

A. 5%

B. 10%

C. 20%

D. 15%

E. 1%

F. None of the above

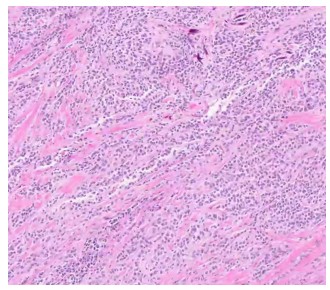

## Cognition Question Example

**Question:** What are the vesicular structures seen at the left and bottom borders? What is their function?

**Options:**

A. Early endosomes which are involved in sorting of endocytosed material.

B. Electron opaque cross-sectioned kinetodesmal fibers for structural support.

C. Granulo-fibrillar material involved in cellular structure.

D. Mitochondria which provide energy to the cell.

E. Axosome which gives rise to microtubules.

F. None of the above.

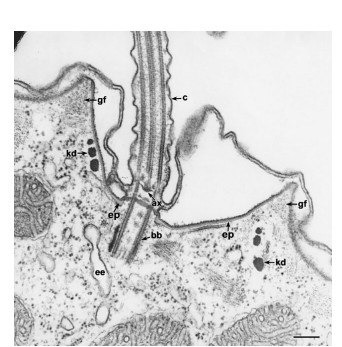

## Cognition Question Example

**Question:** Are synapses visible in this image?

**Options:**

A. Yes, synapses are clearly visible as distinct structures.

B. No, synapses are not visible as the image lacks specific synaptic markers.

C. Yes, synapses are visible where dendrites and axons come into close proximity.

D. No, synapses cannot be identified due to the resolution limitation of fluorescence microscopy.

E. Yes, the orange and gray colored areas clearly show synapses.

F. None of the above.

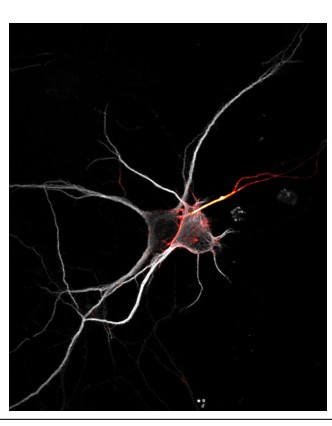

| **Cognition Question Example** |
|---|

**Question:** Describe the spatial relationship between organelles in the cell.
**Options:**
A. Lysosomes are in two populations, one at the cell center and one at the periphery.
B. The Golgi apparatus surrounds the mitochondria while the ER is dispersed in the cytosol.
C. ER and lipid droplets cluster together at the cell periphery.
D. Mitochondria are positioned mainly in the cell periphery, while lysosomes are located centrally.
E. Lysosomes are located close to the nucleus while peroxisomes are scattered throughout the cytoplasm.
F. None of the above.

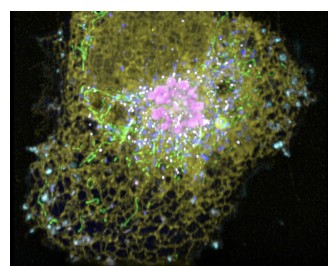

| **Cognition Question Example** |
|---|

**Question:** Count the number of red cells in this image two days and four hours after initial co-culture in a 96 well plate.
**Options:**
A. 7500 cells
B. None of the above
C. 8500 cells
D. 9000 cells
E. 9500 cells
F. 8000 cells

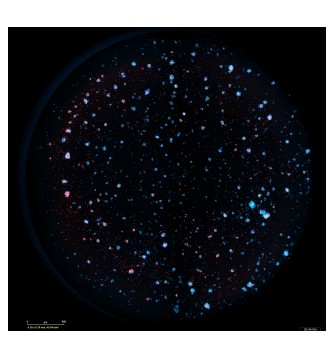