# OpenReview forum: "Micro-Bench: A Microscopy Benchmark for Vision-Language Understanding"
_NeurIPS.cc/2024/Datasets_and_Benchmarks_Track — NeurIPS 2024 Track Datasets and Benchmarks Poster_

### Official Review · Reviewer_YbH8 · 2024-07-08
**Review of μBench**

**Rating:** 3
**Confidence:** 5
**Correctness:** See opportunities for improvement.

**Review:**

In summary, while I agree with the overall concept and motivation behind the authors' proposed dataset, and I appreciate how they've ensembled a diverse suite of microscopy datasets with ample metadata available for each, I have many critiques about its execution. First, many of the tasks seem to be heavily confounded by the source of the dataset and on highly correlated images. Second, many of the VQA tasks do not seem to be solvable from the images alone, are only solvable with heavy assumptions or context, or involve virtually indistinguishable classification labels. Third, the vast majority of tasks do not seem to have any utility or applicability to biomedical research and involve classifying metadata that would be automatically collected during experimental acquisition, or otherwise known to the biomedical expert by default though experimental design. Together, this raises concerns that performance on this dataset may largely be measuring "gaming" behaviors from VLMs (e.g. collapsing to a single modal prediction on a set of highly perceptually similar images), instead of polling for any fundamental understanding of biomedical knowledge in these models.

**Strengths:**

In general, I think the authors have done a lot of work in cataloguing different microscopy datasets and making these resources available as a centralized resource. I also appreciate the authors' intentions in creating this dataset - I agree that more domain-specific evaluations like these have the potential to open up new applications, and here, the motivation behind this dataset is the expansion of VLMs to scientific research, which I think could use more highlight in the community.

**Additional Feedback:**

N/A

**Clarity:**

In general, yes, but I think the authors could do more to be upfront about the limitations of their dataset, as I highlighted in opportunity for improvement.

**Documentation:**

Yes

**Limitations:**

See point 3 in feedback.

**Opportunities For Improvement:**

1) In total, there are only 24 microscopy datasets represented in this effort. While I am loathe to say "only" because I recognize this still does reflect a significant effort in cataloguing microscopy datasets beyond the scale of most public microscopy curation efforts, this raises issues with the validity of the classification tasks - many of the labels (e.g. for subdomain, stain, submodality) are exclusive to one or a few datasets. Perceptually, the images within each dataset are very similar (and this is compounded by the fact that at least 4 of the datasets are synthetic datasets, which tend to be even more limited in their variability): they tend to be of the same resolution, imaging modality, often the same cell type, etc. Together, this creates a risk that performance on the benchmark is largely driven by the fact that images are not independent and independently distributed - because the images are so perceptually similar, you'd expect a VLM to output a similar answer to most questions for most images in most datasets. In other words, I'm not convinced of the claim that performance on this dataset means a VLM necessarily knows what e.g. a "light microscopy" image is, because the model has not been validated on samples representing the full distribution of light microscopy images - at the extreme, you could claim that the effective sample size is only 24 images because of how tightly distributed each dataset is.

2) Much more care should be given to the construction, preprocessing, and curation of labels and images - even at a scan of the HuggingFace database and the sample questions, I can spot many issues. I've listed some of these as subpoints of point 2, but this is not intended to be exhaustive - my concern is that there are a litany of issues present from even a cursory look through curated examples from the authors themselves, that this is indicative of larger systematic issues with the dataset.

2a) Many of the labels are arbitrary, and not classifiable from the images without additional context. For example - what is the difference between "cell biology" and "cell and molecular biology"? From what I can tell, some synthetic fluorescent microscopy datasets are "cell biology" - I guess this is because these datasets are designed to test if models can predict aspects of the morphology of these cells. But stuff like classifying cell cycle stage from brightfield morphological phenotype of cells are coded as "cell and molecular biology" - and there's a separate "cell cycle" category that doesn't include images from this dataset. Similarly, I see images of smooth muscle under "gastrointestinal pathology" - considering this tissue type occurs in multiple different organs, is it even possible to isolate from the crop alone that this intended to understand gastrointestinal pathology (as opposed to pathology in another organ) from the crop alone and no other context?

2b) Many of the questions are not solvable from the images alone even for human experts - there are numerous underlying reasons for this. Some are due to image resolution - for example, on page 55 of the supplementary, a cell counting problem is posed, and it's clear from the context of the question that the original microscopy image was imaged at high resolution (i.e. the correct answer is 8000 cells, and this must be ballparked within a +/- error range of 500 cells). But at an input resolution of 224x224 pixels, most of the cells become too small to even be visible, so counting the cells at this precision becomes impossible (and I suspect some of the pathology examples have similar issues when shown at the WSI-level - e.g. the one in Page 2 of the main text). Some are due to artifacts from applying the entire dataset label to individual images - for example, many of the pathology datasets contain background or debris tiles, and it is impossible to even guess what the micrograph would be used for without any tissue/cell content unless I'm overfitting to the domain of the image (at maximum, I can infer some are H&E images due to the background color, but some datasets appear to be color-corrected). Some are due to unexplained or assumed coloring conventions in fluorescent microscopy images in the context - in page 2 of the main text, I have no way of really knowing that the red in the Cognition Immunology image is really RFP - I don't know what wavelength the red channel was acquired at, nor do I know if this is RFP or an antibody stain colored red etc. Similarly, in page 55 of the supplementary, I don't know what stains each of the colors represent - I can infer that yellow is some kind of cytoskeletal component, the pink is some kind of nuclear component, etc, but typically with the interpretation of these images, I'm given information about what the markers actually were. Without this context, I'm making assumptions that are not scientific - e.g. many punctate compartments look highly similar to each other and can become confounded under perturbation treatments (e.g. if a marker is trafficked between secretory compartments as a result). Confusingly, this is inconsistently applied - e.g. I'm given information about the IHC staining target in the pathology example in Supplementary Page 51... Finally, there seem to be issues with the data itself for some questions? For the colorectal image in Supplementary Page 50, the responses do not indicate a texture classification task. I'm not sure if this is an issue with the dataset itself, or just an error in the supplementary.

3) In general, while this benchmark is framed as interpreting and reasoning about microscopy images, as far as I can tell - the vast majority of questions posed are not useful for biomedical research. This is primarily because the majority of questions are framed around taking metadata that is already automatically known about these images from experimental design and converting them into questions for VQA. As I pointed out examples of above, this creates artifacts where the metadata is not determinable from the images, but it also means in the vast majority of cases, the VLM is simply predicting metadata already known to the experimentalist.

4) The use of a LLM to generate distractor options means there is a huge amount of variability in the difficulty of multiple-choice questions. As I mentioned, the cell counting question in Supplementary Page 55 is unsolvable at the resolution of the images (although if the distractors had coarser grained options, it could be solvable). But in the brightfield cognition question of Supplementary Page 53, what is a challenging open-ended question to answer (i.e. the cancer cell line imaged) is now made trivial, because the distractors are so wildly divergent in morphology from what is imaged that you can figure out what the right answer is by process of elimination. This methodology also seems to oppose the intent of these sets of questions, which the authors explain as curating questions that require reasoning or domain expertise. From some of the examples given in the main text, some of the domain expertise is reporting not just what is in the image but providing some context about the biology of the result. But the coarse-grained nature of some of the distractors mean that the question is now solvable just by reporting image content - e.g. in Supplementary Page 53, the organelle identification answer is framed as needing the model to report both organelle and function correctly, but because all of the other multiple choice questions have different organelles, the function part is now given.

**Relation To Prior Work:**

Yes

**Summary And Contributions:**

Here, the authors release a multi-modal benchmark for microscopy comprising of 22 vision-language modeling tasks and show that current VLMs perform poorly.

---

> ### Author Rebuttal · Authors · 2024-08-18
>
> Thank you for acknowledging the strong concept and motivation as well as the significant effort that went into assembling μBench. We would like to address some potential misconceptions that may clarify your concerns. We have taken the issue of data quality and integrity seriously from the beginning and value this opportunity to present additional experiments and analyses that further reinforce the rigor and validity of μBench. Below, we address each issue in a point-by-point response.
>
> A. Solvability of VQA Tasks
>
> "Many of the VQA tasks do not seem to be solvable from the images alone, are only solvable with heavy assumptions or context, or involve virtually indistinguishable classification labels"
>
> We respectfully disagree
>
> First, we emphasize that μBench takes a holistic approach with 3 components. Two are perceptual:  coarse-grained (e.g., domain, modality, stain) and fine-grained (cell type, subcellular structure, healthy or diseased state etc.). We also have a cognition component that includes questions that require reasoning about images or connecting image features to a broader knowledge base. This holistic approach evaluates VLMs across multiple axes of perception and cognitive tasks. Simple coarse-grained tasks are common, and  Chest X-ray benchmarks for interpretation include “view classification” (i.e., X-ray taken AP, PA, and Lateral).
>
> * Coarse-grained: Context is omitted to avoid trivial question-answer pairs. However, as shown in Table 1, all VLMs significantly outperform random chance on these tasks. Notably, GPT-4o achieves over 90% accuracy for modality, 70% for submodality, 60% for staining technique, and 60% for subdomain questions
>
> * Fine-grained: Tasks are derived from real-world biomedical research, such as red blood cell classification, staging the cell cycle, and grading Pap smears. They are relevant and the original authors solved them with image analysis or ML. We also show these tasks are easily solvable using a logistic regression classifier trained on DINOv2 ViT-S/14 features (Fig 6). We add  UMAP plots of the DINOv2 features for each dataset to show the qualitative separation of classes.
>
> * Cognition: There is a 45% performance gap between random chance and GPT-4o, with a wide range of performance by other general VLMs. We acknowledge these tasks are challenging but appear to be solvable.
>
> External review of feasibility and task difficulty:
>
> We revisit the feasibility of the VQA tasks by recruiting 10 external experts with experience in microscopy and biomedical research (median 9.5 yrs). Experts were diverse and included roles such as post-docs, two board-certified pathologists, and one PI. They determined the perception dataset tasks could be feasible with proper training, although difficulty varies. Average difficulty was 2.5 on a 5-pt Likert scale, with the most challenging tasks rated ~4.3.  We excluded questions identified as challenging from our analysis, including the VQA examples from your review (e.g. debris tiles). We re-computed Table 1 and found no change in VLM rankings, with an average acc. diff. of only 0.4% (Table 6)
>
> B. Perceptual Similarity
>
> "Perceptually, the images within each dataset are very similar"
>
> This can be an issue with any dataset. We compute the perceptual hash using the discrete cosine transform method on the grayscale image to produce a 64-bit binary hash to identify perceptually similar images (see PDF). We use a Hamming distance of ≤5 as a threshold for similarity.
>
> As shown in (Table 1 and 2), the average weighted perceptual similarity of μBench benchmark is no different from other influential biomedical datasets and benchmarks used in hundreds of studies:
>
> MicroBench:13.43%
>
> MedMNIST:13.37%
>
> Patch Camelyon:15.90%
>
>
> We exclude all datasets with high perceptual similarity (>40%) and re-compute the performance metrics to find no change in VLM ranking and the absolute difference in performance is small.
>
> C. Utility
>
> "…vast majority of tasks do not seem to have any utility or applicability to biomedical research and involve classifying [experimental] metadata"
>
> Questions about experimental metadata only include the “coarse-grained perception”, which is 1 of 3 evaluation axes. It does not apply to other axes and hardly qualifies as the “vast majority". See section A for our 3 axes of evaluation. Coarse-grained perception is critical for VLMs to perform higher-level reasoning since it sets a frame of reference. What is normal in one setting could be abnormal in another. That depends, in part, on the context (microscope, specimen, stain, etc). Fine-grained tasks are directly applicable to biomedical research, as described in A. They are biologically meaningful tissue classes, cell types, organelles, disease states, etc, of varying difficulty. The classes or morphologic groups were determined by the original authors from diverse fields. For a further discussion, please refer to our response to 2NDi.
>
> D. Image resolution
>
> “Many of the questions are not solvable from the images alone even for human experts… Some are due to image resolution”
>
> We will release all dataset images at full resolution. VLMs still require images to be resized  (often 224x224) regardless of the original resolution. See the above section on “External review of feasibility and task difficulty”, where experts review tasks and downsampled images to assess task feasibility.
>
> E. Questions about specific VQA or supp. images
>
> Ten experts in section A found tasks challenging, but overall feasible. They identified VQA with challenging or ambiguous answers. We excluded these data from our re-analysis to find a negligible difference and no change in the VLM ranking. The brain tissue image on Page 2 led to a broad claim that many pathology VQA were flawed. To clarify, we consulted two neuropathologists, who confirmed that the image-question-answer triple is solvable at a resolution of 224x224. We clarify other questions about individual examples in the attached PDF.

---

> > ### Comment · Reviewer_YbH8 · 2024-08-28
> > **Response to Authors**
> >
> > Thank you to the authors for their response. In general, I am not fully satisfied that their rebuttal addresses all of the concerns I raised in my original review - I have further questions about the dataset, as well as clarifications about my original review.
> >
> > First, I disagree that global metrics across the entire dataset are sufficient to address my concern about many images being insolvable - to clarify, what I mean is that these tasks being solvable using the biological content of the image, not technical covariates. I agree the fine-grained classification tasks will generally be solvable, because these labels are taken from previous classification datasets. Many of the coarse-grained labels involve propagating meta-data, typically at the dataset level, to each individual instance. As I pointed out in my original review, this causes instances to be labeled inappropriately (e.g. blank histopathology tiles). I'm also concerned about the issue of class collision, because meta-data across different datasets, from my dive into the dataset, has not been sufficiently reconciled (e.g. "cell biology" vs "cell and molecular biology"). This does not rule out that models can still perform well above random chance due to identification of technical covariates (e.g. background staining in empty histopathology tiles).
> >
> > Second, I'm not fully convinced by the perceptual metrics the authors are using to measure similarity. First, converting all images to greyscale images underestimates the covariate shift here, especially as the images are presented to the LLM as RGB images - many of the datasets they integrate have completely different color histograms. Second, presenting the perceptual hash value without a positive/negative control and only in respect to other datasets means it's difficult to understand if this is actually measuring the extent of perceptual similarity in the dataset - I find it concerning that Patch Camelyon, which only uses color images from histopathology, has a relatively similar perceptual hash score to the other two datasets, which integrate both RGB and B/W images from multiple domains. This throws doubt on the metric that they've using - I recommend a more sophisticated metric like FID.
> >
> > Third, I'm encouraged that the authors have reviewed the VQA questions with external experts and carried out their ablation study of the most challenging questions. That being said, this still doesn't address my concern about the methodology used to curate confounders for questions to begin with - the claim of this part of the benchmark is that it assesses reasoning, but as I noted, many of the confounding answers are so divergent from the correct answer that it removes the reasoning challenge. Although this review of VQA questions addresses the issue that some questions may not be answerable - it does not address the issue that the curation of confounders may be reducing the extent this is a "reasoning" benchmark.

---

> > ### Author Rebuttal · Authors · 2024-08-31
> >
> > Thank you for your follow-up questions and for clarifying your original review. Based on your response, it appears that 3 out of the 5  initial concerns have been addressed. To recapitulate:
> >
> > 1) YbH8“I agree the fine-grained classification tasks will generally be solvable” We both agree that the fine-grained perception tasks are solvable.
> >   2) YbH8” I'm encouraged that the authors have reviewed the VQA questions with external experts and carried out their ablation study of the most challenging questions.”  Thank you for recognizing our additional quality control efforts. We agree that the cognition tasks can be solved, but the difficulty varies.
> >   3) YbH8“fine-grained classification tasks [..] [have] labels taken from previous classification datasets”. We believe that μBench will be a valuable and important resource since it reflects tasks pursued in the biomedical research community.
> >
> > We discuss the remaining concerns below:
> >
> > A.2  VQA task solvability
> >
> > YbH8”I disagree that global metrics across the entire dataset are sufficient to address my concern about many images being insolvable”
> >
> > Per our initial response (section A), we use multiple levels of evidence to support  the solvability of μBench and we do not rely on “a single metric to determine task solvability”:
> > 1) Ten external experts review all datasets and VQA tasks, rating difficulty and feasibility (Section  A).
> > 2) We demonstrate that a simple linear classifier achieves high accuracy (Response Table 3).
> > 3) Table 1 shows a wide range of VLM performance across tasks, including high performance by GPT4o.
> > 4) We visualize  DINOV2 UMAP embeddings, demonstrating heterogeneity within a dataset.
> >
> > YbH8“[I disagree that] these tasks [are] solvable using the biological content of the image, not technical covariates”
> >
> > We agree that ”fine-grained classification tasks will generally be solvable”. This concern appears to be limited to the coarse-grained subset. As described previously, coarse-grained tasks evaluate VLM’s capability to interpret or infer the context (i.e., microscope, stain, etc.), setting the stage for reasoning. It is often unclear how to precisely separate “biological content” from “technical covariate”; for example, an H&E tissue image will have a different RGB histogram and texture than a fluorescent or electron microscopy tissue image.
> >
> > YbH8“coarse-grained labels involve propagating meta-data [...]  this causes instances to be labeled inappropriately (e.g. blank histopathology tiles [...] This does not rule out that models can still perform well above random chance due to identification of technical covariates"
> >
> > We reiterate that the metadata propagation only applies to coarse-grained tasks. The perseveration on coarse-grained metadata and “blank histopathology tiles” does not acknowledge our analyses, removing these problem VQA questions (and others with potential issues) to show no significant change in VLM ranking (section A).
> >
> > Secondly, it is inaccurate to broadly claim that all coarse-grained metadata is wrong and misrepresents our approach. For example, a brightfield microscopy image is markedly different from an electron or fluorescence image, and using that metadata would be appropriate.
> >
> > Class collision:
> >
> > We have already updated the dataset to combine “cell biology” and “cell and molecular biology” classes. Our apologies if this was not clear earlier.
> >
> > B.2 Perceptual Similarity:
> >
> > B2.1 YbH8“I'm not fully convinced by the perceptual metrics the authors are using to measure similarity.”
> >
> > We used a perceptual hashing algorithm (pHash) to address YbH8’s original concern:
> >
> > B2.1 YbH8“Perceptually, the images within each dataset are very similar.”
> >
> > The pHash using the DCT was not a custom metric. This is a well-established and widely accepted algorithm used in image retrieval and similarity searches for over 10 years (Longjiang et al. 2006, Tang et al. 2014, and Li et al. 2018). It is still used today for image search and retrieval and copyright infringement, among other use cases.
> >
> > YbH8“I find it concerning that Patch Camelyon, which only uses color images from histopathology, has a relatively similar perceptual hash score to the other two datasets”
> >
> > This may be a misunderstanding of pHash. The original claim (B2.1) argues similarity of “images within each dataset”. Thus, for each image, we compute a pHash, and then find images with a similar pHash. Table 1 shows the fraction of “similar images” (HD<5) within a single dataset to address B2.1.
> >
> > Further, we clarify that pHash uses the DCT and operates in the frequency domain. A review of the pHash similar images in PCAM shows many near duplicates (HD=0). The PCAM creator used an unspecified balancing strategy to ensure a 50:50 class ratio. In contrast, we show that images with a close pHash (HD<5) from μBench data are visually distinct.
> >
> > YbH8“I recommend a more sophisticated metric like FID”
> >
> > FID compares the distributions of two image datasets, not individual image pairs, making it unsuitable for detecting perceptually similar images within a dataset. FID's focus on distributional differences lacks the granularity required for precise pairwise image comparisons. The DCT-based perceptual hashing algorithm we used is designed to identify similar images, providing a more accurate and efficient solution.
> >
> > G Confounders in Cognition subset:
> >
> > YbH8 “the curation of confounders may be reducing the extent this is a reasoning benchmark.”
> > Cognitive tasks span multiple difficulties and different levels of reasoning. All VQA-MC will have some options that are easier to rule out than others, but this does not imply the dataset is no longer “a reasoning benchmark”. If anything, the easy confounders should make the tasks more solvable, contrasting with the poor quantitative performance on the cognition subset.
> >
> > We appreciate your input, and our additional analyses in response to questions have improved our work. Thank you for engaging with us during the discussion period.

---

> > > ### Author Response · Authors · 2024-08-31
> > > **References**
> > >
> > > [1] Longjiang Yu and Shenghe Sun. Image robust hashing based on dct sign. In 2006 International
> > > Conference on Intelligent Information Hiding and Multimedia, pages 131–134. IEEE, 2006.
> > >
> > > [2] Tang, Z., Yang, F., Huang, L., & Zhang, X. (2014). Robust image hashing with dominant DCT coefficients. Optik, 125(18), 5102–5107. doi:10.1016/j.ijleo.2014.05.015.
> > >
> > > [3] Li, Zhongyu, et al. "Large-scale retrieval for medical image analytics: A comprehensive review." Medical image analysis 43 (2018): 66-84.

---

### Official Review · Reviewer_jGSc · 2024-07-17
**Microscopy benchmark for vision-language understanding**

**Rating:** 7
**Confidence:** 3
**Correctness:** Preprocessing steps and experiments s…
**Clarity:** The paper is clearly written, though …

**Review:**

Benchmarks made only of a combination of open datasets are not really original, but authors stand out by providing expert text annotations for evaluation of VLMs. That is a lot of work and definitely a significant contribution for microscopy community.
The paper is clearly written and supplementary includes a lot of details.

**Strengths:**

- Authors collected a dataset of diverse microscopy modalities.
- Many VLMs are benchmarked and compared with this dataset
- Supplemental material is rich in details, such as:
    - Examples of questions and captions for each source dataset
    - Examples of image-text pairs

**Additional Feedback:**

The discussion on catastrophic forgetting reminded me about another paper on weights in LLMs (general vs fine-tuned) and prior work related to that paper https://arxiv.org/pdf/2302.04863 . Do you think insights from such prior work can be somehow incorporated into yours?

**Documentation:**

The Dataset and GitHub link are buried in the supplementary (at the time of the review was not public) - should be in abstract in the final version.

**Ethics:**

I see no ethical concerns related to presentation of this paper, the datasets are openly available and only some of the image were originally related to human subjects. Authors also have a discussion section regarding ethics.

**Limitations:**

Authors describe limitations in the supplementary, specifically the dataset size for certain modalities, that could be further expanded if the benchmark will be supported in future.

**Opportunities For Improvement:**

Lines 620-622: Inclusion of the pollen dataset. Inclusion of this dataset makes the benchmark not as specialised and I am not sure it serves well as it is a completely different domain. If one is interested in performance related to cell microscopy, why would one need additional data domains? Benchmarks should contain diverse data (in terms of data sources, batches, microscopy modalities, maybe species) but still focus on one thigh.

To me, the paper seems to lack an extended discussion/conclusion in the main text.

Typos :
- Line 128 VAQ -> VQA
- Line 154: end of the sentence µ-Bench
- 872 in supplementary - repeated ‘table’

**Relation To Prior Work:**

The related work is clearly described and referenced. Authors describe why the proposed benchmark stands out.

**Summary And Contributions:**

The paper offers a benchmark for evaluation vision-language benchmarks to evaluate generalist or specialist VLMs in microscopy image interpretation. The benchmark aims to evaluate the ability of both object recognition (perception) and reasoning (referred to as cognition in the paper) of popular general-use and specialized VLMs.

---

> ### Author Rebuttal · Authors · 2024-08-18
>
> We sincerely appreciate your detailed and thoughtful review of our manuscript, "μ-Bench: A Vision-Language Benchmark for Microscopy Understanding." We are encouraged by the recognition of the value that our contributions and the significant effort involved in creating a comprehensive benchmark for evaluating vision-language models (VLMs) in microscopy. Below, we address each of your points in detail.
>
> * R2 “Lines 620-622: Inclusion of the pollen dataset. The inclusion of this dataset makes the benchmark not as specialized and I am not sure it serves well as it is a completely different domain. If one is interested in performance related to cell microscopy, why would one need additional data domains? Benchmarks should contain diverse data (in terms of data sources, batches, microscopy modalities, maybe species) but still focus on one theme.”
>
> Thank you for your observation. We understand your point about including the pollen dataset potentially making the benchmark less specialized. While cell microscopy images constitute a significant portion of our dataset, we also incorporate images of bacteria, viruses, diverse model organisms, and single-molecule imaging to represent the diversity of biological and biomedical research. Pollen is an allergen studied across multiple disciplines, such as botany, environmental sciences, and immunology. We believe including a portion of diverse data, such as pollen, enhances the benchmark by broadening its applicability to more biology disciplines and may facilitate testing generalization capabilities across different microscopy domains. Also, we hope that incorporating diverse data, such as pollen, will foster a sense of inclusion in the benchmark that will encourage diverse microscopy researchers to contribute to future versions.
>
> However, we acknowledge the importance of maintaining focus within a benchmark. To address this, we offer highly annotated HuggingFace arrow and parquet datasets that allow researchers to filter the dataset by specific criteria such as microscopy domain, subdomain, modality, submodality, staining, and more. This flexibility ensures that users can tailor the benchmark to focus only on specific areas of interest, such as cell microscopy, while benefiting from the broader dataset. We have a demo Colab notebook in our GitHub repository highlighting this dataset filtering.
>
> * R2”The discussion on catastrophic forgetting reminded me about another paper on weights in LLMs (general vs fine-tuned) and prior work related to that paper https://arxiv.org/pdf/2302.04863. Do you think insights from such prior work can be somehow incorporated into yours?”
>
> 2A) Thank you for bringing this relevant work to our attention. We have reviewed the suggested paper and incorporated it into our discussion section. Specifically, we discuss how exploring regions between models can yield new models that perform comparably or even better than those obtained via fine-tuning, including on tasks not previously fine-tuned. This is particularly relevant given the challenges of curating high-quality, specialized microscopy data at the same scale as general Internet-based datasets (e.g., LAION 5B). Consequently, domain-specific biomedical VLMs often fine-tune a general-purpose model, which is a good practice. However, biomedical VLMs often overlook strategies to mitigate catastrophic forgetting, as highlighted by the fact that the specialist biomedical VLMs we evaluated did not initially employ such strategies. Overall, this expanded discussion emphasizes the importance of adopting strategies to mitigate catastrophic forgetting when fine-tuning VLMs for specialized biomedical applications.
>
> We have compiled details from our extended discussion and included them in the supporting PDF document for your review.
>
> * R2 “To me, the paper seems to lack an extended discussion/conclusion in the main text.”
>
> We appreciate your feedback regarding extending the discussion and conclusion. We have revised the manuscript to expand the discussion section, providing deeper insights into our findings, particularly regarding the implications of catastrophic forgetting in fine-tuned models (2A). This was an excellent suggestion, and the expanded discussion and conclusion strengthened the manuscript.
>
> * R2”Typos and Minor Corrections: Line 128: VAQ -> VQA,  Line 154: end of the sentence µ-Bench Supplemental page 872: repeated ‘table’”
>
> Thanks for pointing out these minor oversights. We have corrected the highlighted typos and repeated words. In addition, we performed another round of copyediting on the text to ensure there were no other issues.
>
>
> * R2”The Dataset and GitHub link are buried in the supplementary (at the time of the review was not public) - should be in abstract in the final version.”
>
> We have added the dataset and GitHub links to the abstract to make them easily accessible to readers.
>
> * R2” I see no ethical concerns related to the presentation of this paper. The datasets are openly available, and only some of the images were originally related to human subjects. The authors also have a discussion section regarding ethics.”
>
> Thank you for recognizing our attention to ethical considerations. We are committed to prioritizing the ethical curation and processing of public biological samples, including de-identified human tissue. We will continue to maintain this focus as we improve upon the benchmark in the future.
>
> Your thoughtful review and insightful comments helped us refine our manuscript further. These revisions enhance the clarity and impact of our contributions. Please feel free to reach out if you have any further questions or comments. We would be happy to discuss them further!

---

> > ### Comment · Reviewer_jGSc · 2024-08-28
> >
> > Thank you for clarifications and answers.

---

### Official Review · Reviewer_Fyh1 · 2024-07-25
**A VLM Microscopy Benchmark**

**Rating:** 8
**Confidence:** 4
**Correctness:** Yes.
**Clarity:** Very clear and easy to follow.

**Review:**

- **A comprehensive benchmark dataset**: The authors introduce a diverse and extensive benchmark for evaluating vision-language models (VLMs) in microscopy that covers both perception and cognition tasks, providing a holistic evaluation framework for microscopy understanding.

- **Multiple levels of evaluation**: This benchmark has structured evaluation that assesses both perception and cognition capabilities of VLMs for coarse-grained perception tasks, fine-grained perception, cognition and localization.

- **Model insights**: Interestingly, current models fail on all categories, including specialist VLMs trained on domain specific data. Also, fine-tuning can cause catastrophic forgetting. The authors provide a solution to mitigate this i.e., weight interpolation of different models to produce ensemble models that show notable gains in tasks that were otherwise hard.

**Strengths:**

- Addresses a critical gap in the evaluation of VLMs for microscopy.
- Very comprehensive coverage of various microscopy tasks and modalities.
- Rigorous evaluation methodology and provides valuable insights into current model limitations.
- Open-source release promotes further research.

**Additional Feedback:**

Overall a well written paper with a significant research effort in creating this benchmark.

**Documentation:**

Details included.

**Ethics:**

N/A.

**Limitations:**

- Explore methods to mitigate potential biases in the dataset. Did the authors consider including the inherent biases as part of the evaluation.
- In a similar sense, edge cases should be considered in the evaluation. The authors acknowledge and discuss this in the supplementary.

**Opportunities For Improvement:**

N/A.

**Relation To Prior Work:**

Very detailed.

**Summary And Contributions:**

μ-Bench is a comprehensive benchmark for evaluating vision-language models (VLMs) in microscopy understanding. It addresses the lack of standardized, diverse, and large-scale benchmarks in this domain. The dataset encompasses 22 biomedical tasks across various scientific disciplines, microscopy modalities, scales, and organisms. The authors evaluate state-of-the-art biomedical, pathology, and general VLMs on μ-Bench, revealing significant performance gaps and challenges in microscopy image interpretation.

---

> ### Author Rebuttal · Authors · 2024-08-17
>
> We sincerely appreciate your thorough review and positive feedback on our submission, ‘μBench: A Vision-Language Benchmark for Microscopy Understanding.’ We are encouraged that you found our benchmark “addresses a critical gap in the evaluation of VLMs for microscopy” and recognized the “comprehensive coverage” of microscopy tasks/modalities. We devoted significant time and effort to the design, implementation, and evaluation of μBench, and your positive remarks about the “rigorous evaluation methodology” affirm our efforts. You also raised several good suggestions for how we can improve the benchmark. Below, we respond to each of your points in detail.
>
> * R3”The authors introduce a diverse and extensive benchmark for evaluating vision-language models (VLMs) in microscopy that covers both perception and cognition tasks, providing a holistic evaluation framework for microscopy understanding.”
>
> Thank you for recognizing the comprehensive nature of μ-Bench. Our primary goal was to develop a benchmark that offers a thorough evaluation framework for both perception and cognition tasks in microscopy. This holistic approach is essential for capturing the diverse challenges of microscopy image understanding.
>
>  * R3”This benchmark has structured evaluation that assesses both perception and cognition capabilities of VLMs for coarse-grained perception tasks, fine-grained perception, cognition, and localization.”
>
> We appreciate your acknowledgment of the structured evaluation methodology we employed. By incorporating multiple levels of evaluation—coarse-grained perception, fine-grained perception, cognition, and localization—we aimed to provide a detailed assessment of VLMs' strengths and weaknesses. This multi-faceted approach ensures that μ-Bench offers a comprehensive understanding of VLMs' capabilities across various microscopy contexts.We hope this benchmark will prove useful to researchers as they begin to explore specialized domains such as microscopy.
>
> * R3”Interestingly, current models fail on all categories, including specialist VLMs trained on domain specific data. Also, fine-tuning can cause catastrophic forgetting. The authors provide a solution to mitigate this, i.e., weight interpolation of different models to produce ensemble models that show notable gains in tasks that were otherwise hard.”
>
> Thank you for highlighting this important observation. The significant performance gaps and challenges revealed by our benchmark provide crucial insights into the limitations of current models and areas for further research. We are glad that our proposed solution of weight interpolation to create ensemble models was noted as a valuable contribution. We believe this approach can help mitigate catastrophic forgetting and improve model performance on challenging tasks.
>
> * R3”Explore methods to mitigate potential biases in the dataset. Did the authors consider including the inherent biases as part of the evaluation?”
>
> We appreciate this suggestion and acknowledge the importance of addressing potential biases in the dataset. We recognize that bias can be introduced in many different ways. In our discussion of ethics, we acknowledge that human tissue and cell lines are overrepresented compared to cells from other organisms. This includes the ubiquitous use of the HeLa cell line in biomedical research, which was derived from patient tissue without consent or permission. The dataset is also biased toward light and fluorescence microscopy images compared to other techniques. These biases in data distribution partly reflect the usage patterns in biomedical research and data availability
>
> Prioritizing diversity, equity, and inclusion and working to mitigate bias are critical for the safe adoption of AI in biology and biomedical research. During the curation process, we took special care to prioritize diverse samples and record demographic details. We scoured supplementary documents and tables to curate patient information, if available. Detailed metadata was unavailable in many cases, but we made our best efforts. Datasets of human cell lines were annotated with the age, sex, and demographics as recorded in public databases (cellosaurus, cellontology, etc.). We have added the table of demographics to the appendix, which provides transparency on the dataset's coverage. We anticipate this metadata can serve as a tool for researchers investigating how to detect and mitigate AI bias. In future research, we plan to incorporate specific metrics to evaluate for systematic biases.
>
> * R3”In a similar sense, edge cases should be considered in the evaluation. The authors acknowledge and discuss this in the supplementary.”
>
> Thank you for pointing this out. We agree that edge cases can provide critical insights into the limitations of current models. To better address this, we conducted a survey with four biologists who reviewed randomly sampled images from each dataset and task within μ-Bench. They rated task feasibility and difficulty, revealing a distribution that reflects the varying complexity of tasks. This metadata will be integrated into each dataset, helping to contextualize the difficulty of individual tasks and potentially highlighting edge cases. We believe this will provide valuable context for understanding model performance across different scenarios.
>
>
> We appreciate your insightful feedback. If you have any further questions or comments, please let us know.

---

> > ### Comment · Reviewer_Fyh1 · 2024-08-31
> > **Update**
> >
> > I would like to thank the authors for carefully considering my initial concerns, all of which have been addressed adequately following the review of supporting document. I will maintain the score, and still of the view that this work is worth presenting at NeurIPS.

---

### Official Review · Reviewer_2NDi · 2024-07-28
**A thorough and thoughtful benchmark for the use of vision language models in biomedical research.**

**Rating:** 9
**Confidence:** 5
**Correctness:** The reviewer has no concerns regardin…
**Clarity:** Yes.

**Review:**

There will be no shortage of papers claiming to benchmark vision language models in various fields at this conference. This work is necessary and valuable to the community - both in advancing current state-of-the-art models and making them useful to scientists in these domains. The rigor and thoughtfulness of this paper helps it stand apart. It is well-written, well-sourced, and clearly communicates the challenges ahead of both specialist and generalist vision language models (VLMs) in biomedical research.

The authors introduce a benchmark dataset that is varied enough in composition to rigorously test a wide range of VLMs in multiple tasks of interest to biomedical researchers but no so broad as to make these benchmarks shallow or easy to saturate. The effort is well-organized (into perception and reasoning/cognition) and includes a large amount of human intervention. As dataset construction increasingly involves the use of generative models, the inclusion of domain experts at multiple stages of the pipeline is very welcome. I also appreciate the conversion of open-ended questions into multiple choice so as to decompose the problem and avoid LLM-based evaluation of the results.

The other main contribution of the this paper is its characterization of multiple models using this benchmark. A wide variety of models were selected for test and the selection rationale is well-explained. The juxtaposition of generalist and fine-tuned models demonstrates the utility of the benchmark and could provide insight to those hoping to improve the performance of generalist models in the biomedical research domain.

**Strengths:**

The authors do an excellent job of sourcing data, respecting the original licenses, and improving the value to the community by embedding ontological information and providing additional expert annotations. It is well organized by task and provides sufficient depth in each task to robustly evaluate vision language model. Figure 3 does an excellent job communicating the data splits by domain, modality, and task.

The paper also highlights several state of the art models (both generalist and fine-tuned) and organizes them intuitively into contrastive and auto-regressive models. The shortcomings of these models on these tasks for these domains is expected but well illustrates the utility of benchmark going forward. The radar plots in Figure 4 do a job of adapting existing means of visualizing VLM performance to the domains of interest in this paper.

In short, this paper will be of immediate use to the biomedical research community - guiding those seeking to select a model for current experiments/tasks. It will also benefit the broader ML community as they seek to improve fine-tuning methodologies and VLMs broadly. The clarity of the writing helps bridge the gap between these specialties and makes a potentially daunting task easily digestible.

**Additional Feedback:**

N/A

**Documentation:**

Yes.

**Ethics:**

No ethical concerns

**Limitations:**

The authors have adequately addressed the limitations and impact of their work.

**Opportunities For Improvement:**

As mentioned, I think this benchmark will well serve the community in a number of ways. However, I think the mechanics of why a biomedical researcher would want to "talk with their data" could be better elucidated. For instance, an individual researcher likely knows the modality of their data, what the significance of certain images are, what is happening in an image, etc. There is a scale argument to be made here and we have to start somewhere in terms of measuring performance. But the field would benefit from a more in-depth look at how useful these models could be when capable of improved reasoning.

To that end, it may be useful to add to the related work section some context on how current VLMs are being benchmarked in the broader ML community, especially with respect to VQA and reasoning.

**Relation To Prior Work:**

The authors have well positioned the current work against current research in both VLMs and biomedical research.

**Summary And Contributions:**

The authors introduce an expert-curated benchmark to evaluate vision language models (VLMs) across multiple scientific disciplines, modalities, scales, organisms, and states. This benchmark is used to characterize state-of-the-art models (both general and fine-tuned). The authors do a good job of providing context for the current benchmark and conceptually organizing the models under test, which will be of use to readers who may not be familiar with the broader challenges of VLMs but want to make use of them in their specific domain. The perception and cognition datasets will be a welcome addition to the community. They thoughtfully organize a large amount of disparate open data, include ontological information for captioning, and have been enriched by a high-level of human involvement (as opposed to being fully LLM-generated).

---

> ### Author Rebuttal · Authors · 2024-08-17
>
> Thank you for your thorough review, insightful comments, and constructive feedback. We are glad that you recognize “[t]his work is necessary and valuable to the community value”, and that you appreciate our expert-curated benchmark for evaluating vision-language models (VLMs) across diverse microscopy imaging disciplines. We also appreciate your acknowledgment of the context provided and the conceptual organization of the models to help readers understand the broader VLM challenges. The positive remarks about our “rigor and thoughtfulness”, including human involvement and expert annotations, affirm our efforts; highlighting that our  contribution “benefit[s] the broader ML community as they seek to improve fine-tuning methodologies and VLMs broadly”. We are committed to addressing the opportunities for improvement that you have raised to better serve the scientific and machine-learning communities.
>
> *  R1“To that end, it may be useful to add to the related work section some context on how current VLMs are being benchmarked in the broader ML community, especially with respect to VQA and reasoning”
>
> We appreciate the suggestion. We have added a new section, “Benchmarking Vision-Language Models,” at the beginning of the Related Work section. This new section highlights the trend of reusing existing datasets and providing comprehensive multi-level evaluations to “dissect” failure modes in VLMs in a standardized and systematic manner, and to “aid the community in better understanding AI technology and influencing its development.” We hope this addition provides context on current VLM benchmarks and clarifies our motivation for a multi-level benchmark.
>
> * R1“As mentioned, I think this benchmark will well serve the community in a number of ways. However, I think the mechanics of why a biomedical researcher would want to "talk with their data" could be better elucidated. For instance, an individual researcher likely knows the modality of their data, what the significance of certain images are, what is happening in an image, etc. There is a scale argument to be made here and we have to start somewhere in terms of measuring performance. But the field would benefit from a more in-depth look at how useful these models could be when capable of improved reasoning.”
>
>
>
> Thank you for this feedback. We would first like to emphasize that coarse-grained perception tasks (e.g., domain, subdomain, modality, submodality, stain) are primarily for a holistic approach to evaluate VLMs across multiple axes of coarse and fine perception and cognitive tasks. Including simple coarse-grained tasks is common, and benchmarks for Chest X-ray interpretation include tasks such as “view classification” (i.e. whether the X-ray was taken AP, PA, and Lateral). Furthermore, it is possible that in some cases, the person performing biomedical image analysis is not the experimentalist. This is especially true for techniques in cryo-electron microscopy, where computational analysis may be performed by non-biologists. In these cases, a biomedical VLM could answer basic questions regarding image interpretation, artifacts, or basic biology context.
>
>
> We agree with your suggestion to make our argument clearer. We expanded the introduction and discussion to explain how  and why a biomedical researcher would want to "talk with their data." We included scenarios such as democratizing the ability to interpret microscopy outside of one’s specialty, text/chat guided image analysis, determining if image findings have been reported in the literature, facilitating large-scale image annotation, or brainstorming new hypotheses based on image findings, along with case studies demonstrating the utility of improved reasoning capabilities in complex biomedical datasets. We hope these additions provide an in-depth landscape of “these models’ capabilities when reasoning is improved,” motivating our multi-level evaluation, as before an AI system can reason about scientific images, it must accurately interpret them.
>
> Again, Thank you for your detailed comments and insightful questions about the broader context. Your feedback has been invaluable in improving our paper. If you have any further questions or comments, please let us know, we're happy to discuss them further!

---

### Decision · Program_Chairs · 2024-09-26

**Decision:**

Accept (Poster)

**Comment:**

This paper has very mixed scores, the merits have been observed by all reviewers, agreeing that the research scope itself has a large impact. However, one reviewer has serious concerns about this paper, given the impact if this paper is published at NeurIPS 2024 D&B Track. After carefully checking all comments and rebuttals, I also find that some metrics used by the authors are not clearly explained in the paper. Converting all images to greyscale images indeed underestimates the covariate shift, actually, this procedure will change much information contained in the images already. Although it has been seen that some papers used a similar way, it does not mean it should not be clearly explained, especially as the images are presented to the LLM as RGB images - many of the datasets they integrate have completely different color histograms. Given the remaining concerns and the potential impact of this paper on this subfield, we encourage the authors to fully take care of these reviews and try to avoid these concerns in the next version.

Notes from PC: the SAC championed the paper for acceptance.